

# An integrated open-coastal biogeochemistry, ecosystem and biodiversity observatory of the Eastern Mediterranean. The Cretan Sea component of POSEIDON system.

George Petihakis[1], Leonidas Perivoliotis[1], Gerasimos Korres[1], Dionysis Ballas[1], Constantin Frangoulis[1], Paris Pagonis[1], Manolis Ntoumas[1], Manos Pettas[1], Antonis Chalkiopoulos[1], Maria Sotiropoulou[1], Margarita Bekiari[1], Alkiviadis Kalampokis[1], Michalis Ravdas[1], Evi Bourma[1], Sylvia Christodoulaki[1], Anna Zacharioudaki[1], Dimitris Kassis[1], Emmanuel Potiris[1], George Triantafyllou[1], Kostas Tsiaras[1], Evangelia Krasakopoulou[1], Spyros Velanas[1], Nikos Zisis[1]

[1]Institute of Oceanography, Hellenic Centre for Marine Research, Heraklion, 72100, Greece

*Correspondence to*: Frangoulis Constantin (cfrangoulis@hcmr.gr)

**Abstract.** There is a general scarcity of oceanic observations that concurrently examine air-sea interactions, coastal-open ocean processes, and biogeochemical (BGC) parameters, in appropriate spatiotemporal scales, and under continuous, long-term data acquisition schemes. In the Mediterranean Sea, the resulting knowledge gaps

and observing challenges increase, due to its oligotrophic character, especially in the eastern part of the basin. The oligotrophic open Cretan Sea's biogeochemistry is considered to be representative of a greater Mediterranean area up to $10^6$ km$^2$, and understanding its features may be useful on even larger oceanic scales, since the Mediterranean Sea has been considered a miniature model of the global ocean. The spatiotemporal coverage of BGC observations in the Cretan Sea has progressively increased over the last decades, especially since the creation of the POSEIDON

observing system, which has adopted a multiplatform-multiparameter approach, supporting BGC data acquisition. The current POSEIDON system's status includes open and coastal sea fixed platforms, a Ferrybox (FB) system, and Bio-Argo autonomous floats, that deliver remotely Chlorophyll-a (Chl-a), O$_2$, pH and pCO$_2$ data, as well as BGC-related physical parameters. Since 2010, the list has been further expanded to other BGC (nutrients, vertical particulate matter fluxes), ecosystem and biodiversity (from viruses up to zooplankton) parameters, thanks to the

addition of sediment traps, frequent R/V visits for seawater-plankton sampling, and of an ADCP delivering information on macrozooplankton-micronekton vertical migration (in the epi-, mesopelagic layer). Gliders and drifters are the new, currently under integration to the existing system, platforms, supporting BGC monitoring. Land-based facilities, such as data centers, technical support infrastructures, calibration laboratory, mesocosms, support and give added value to the observatory. The data gathered from these platforms are used to improve the

quality of the BGC-ecosystem model predictions, which have recently incorporated atmospheric nutrient deposition processes and assimilation of satellite Chl-a data. Besides addressing open scientific questions at regional and international level, examples of which are presented, the observatory provides user oriented services to marine policy-makers and the society, and is a technological test bed for new and/or cost-efficient BGC sensor technology and marine equipment. It is part of European and international observing programs, playing key role

in regional data handling and participating in harmonization and best practices procedures. Future expansion plans consider the evolving scientific and society priorities, balanced with sustainable management.



# 1    International need for observatories

Oceans are complex dynamic systems embracing various physical, chemical and biological processes interacting on a wide range of time and space scales. The increasing anthropogenic pressures add another layer of complexity to their study. Ocean observatories are long-term infrastructures dedicated to multiple in situ observations (from air-sea interface to the ocean bottom and the water column), which are maintained over long timescales, with adequate temporal resolution, and are designed to address interdisciplinary objectives over wide spatiotemporal scales. The observatory concept created by scientific needs is driven today by societal needs to understand the connections between ocean processes and society (Ruhl et al., 2011), with the necessity for continued long-term ocean observation also recognized politically by the international community by the United Nations Conference on Sustainable Development at Rio+20 (Cicin-Sain et al., 2011). At the European level, the coordinating bodies (European Marine Board and EuroGOOS) of the European observatories, besides addressing specific scientific questions, come also to address Europe's societal and policy demands for sustainable use of the seas, ecosystem-based management, and establishment of environmental status indicators, as expressed by the EU's Marine Strategy Framework Directive (MSFD).

Biochemical-ecosystem research is a key component in ocean observatories. The International Geosphere-Biosphere Programme (IGBP) (leaded by the Intergovernmental Oceanographic Commission of UNESCO), since its first projects (JGOFS - Joint Global Ocean Flux Study) as well as with its current ones (IMBER - Integrated Marine Biosphere Research) has realized that key biogeochemical parameters and ecosystems must also be systematically and long-term observed, in order to study marine biogeochemical cycles and their interactions with the ecosystems, at the seasonal and decadal scale, as well as in short-term episodic events (www.imber.info).

Biogeochemical-ecosystem long-term data are, however, available only from few locations worldwide (Karl, 2010) and scarcity increases when considering only open (deep) ocean observatories (Ruhl et al., 2011). Recently the European Marine Board (EMB) has published a position paper on critical challenges of deep-sea research, stressing that a current limitation of observatories is that they mainly monitor almost exclusively abiotic variables (Rogers et al., 2015).

# 2    A strategic location to study the unknowns of the Eastern Mediterranean

The Mediterranean Sea also "suffers" by this scarcity of observatories, especially with an open ocean component. Although long term time-series of physical, chemical and plankton ecosystem parameters exist in several shallow (<200 m) coastal sites (<10 n.m. offshore) (Berline et al., 2012; Eker-Deverli et al., 2006; Goffredo and Dubinsky, 2014), few offshore and deep ocean (>1000 m) biogeochemical observatories exist in the Mediterranean (Ruhl et al., 2011; www.eurosites.info). This leaves several questions for its physical and biogeochemical status unanswered, including among others: processes occurring in intermediate and deep layers, variability of the stock of nutrients, biological pump functioning, mesopelagic community structure, multiple stressor impact on ecosystem functioning (Malanotte-Rizzoli et al., 2014).

In the open Eastern Mediterranean Sea studies of biological processes have been focused on various spatial scales (large to meso-scale) in numerous campaigns (POEM, PELAGOS, MATER, INTERREG-N.AEGEAN, METROMED, ANREC, KEYCOP, SESAME, BOOM etc); however these studies were at the best



seasonal. Only two studies on the annual cycle of biochemical parameters were performed [Cretan Sea (Tselepides and Polychronaki, 2000) and Cilician Basin (Eker-Develi et al., 2006)], however with no data on micro- and mesozooplankton. It is worth noticing that there are very few interannual scale studies of mesozooplankton in the open Mediterranean Sea (Siokou-Frangou et al., 2010).

5   The Cretan Sea biogeochemical – ecosystem observatory presented here, comes to fill the above gap. This observatory is located in the most voluminous and deep (2500 m) basin of the Aegean Sea, giving the opportunity to study not only properties limited to the specific area, but also the characteristics of the open Eastern Mediterranean basin. In fact, the open Cretan Sea's biochemistry has been estimated that it represents an area in the Eastern Mediterranean of 0.6-1.6 x $10^6$ km$^2$ depending on the parameter (Fig. 1) (Henson et al., 2016).

10 Understanding Cretan's seas' features may be useful on larger oceanic scales, since the Mediterranean Sea has been considered a "miniature ocean" that can be used as a model to anticipate the response of the global ocean to various kinds of pressures (Bethoux et al., 1999). These Cretan Sea features, presented below, underline the strategic location of the observatory.

### 2.1 Coupling of biochemistry with circulation patterns

15   The Cretan Sea plays an important role in the dynamics of the Eastern Mediterranean circulation and has been considered as heat, salt, and dissolved oxygen reservoir, since high temperature (>14 °C), salinity (>38.9) and dissolved oxygen concentrations (>4.3 ml/l) were detected in its intermediate and deep layers (Theocharis et al., 1993; Souvermezoglou and Krasakopoulou, 2000; www.mongoos.eu/data-center).

   The Cretan Sea is an area of intermediate and/or deep-water formation dominated by multiple scale
20 circulation patterns and intense mesoscale variability (Theoharis et al., 1999; Georgopoulos et al., 2000). Such areas of water formation are key locations for monitoring of the Mediterranean biochemical functioning (Malanotte-Rizzoli et al., 2014). The wintertime convective mixing of the water column and the exchanges of water and mass (diluted, suspended or near-bed) with the adjacent Levantine and Ionian Seas through the straits of the Cretan Arc, make the Cretan Sea the poorer in nutrients and the richer in oxygen, among the principal basins
25 of the Mediterranean Sea (Souvermezoglou and Krasakopoulou, 2000).

   In the late 1980s - early 1990s, the Cretan Sea has become the major contributor of deep/bottom, warmer, more saline water to the Eastern Mediterranean (review by Laskaratos et al., 1999 and references therein), which is considered to be part of a salinity-driven alternating mechanism of the Eastern Mediterranean (Theoharis et al., 2014). This transition known as the Eastern Mediterranean Transient (EMT) alternated dramatically the physical
30 and biochemical properties at the intermediate and deep layers of the whole basin (Theoharis et al., 2009, Roether et al., 1996, 2007, among others). The years following the EMT onset, the intermediate layer of the entire Cretan Sea was occupied by the 'nutrient-rich, oxygen-poor' Transition Mediterranean Water (TMW) that has intruded in the basin compensating the Cretan Deep Water (CDW) outflow (Souvermezoglou et al., 1999). The deep-water outflow from the Aegean/Cretan is also a recurrent phenomenon that has been repeated in the past (i.e. 70's). The
35 concentrations of nutrients within the TMW layer were often two times higher than those observed in the same area during previous studies undertaken before 1992 (Souvermezoglou et al., 1999). After 1995 the chemical characteristics of TMW were less pronounced, a fact probably related to the decay of the transient behavior of the Eastern Mediterranean. Between 1999 and 2002 the Adriatic Sea has gradually started to regain its dense water formation role in the Eastern Mediterranean and, finally, the Adriatic Sea produced dense water in 2012. However,



data collected from different platforms in the Cretan Sea during the 2000s present evidence of gradually increasing salinity in the intermediate and deep intermediate layers after the middle of the decade. During the last decade, dense water formation conditions have been identified showing that the basin is in the process of slowly returning to the state of the late pre-EMT period (Velaoras et al., 2014; Cardin et al., 2015).

The deep-water mass formation in the Cretan Sea has the potential to transfer $CO_2$ into the deep layers. However, there is a sparseness of carbonate system data ($C_T$, $A_T$, pH and $pCO_2$) in the wider Eastern Mediterranean, despite an increasing effort to study these properties in the last decades (Sisma-Ventura et al., 2016). A "winter-time" (February 2006) snapshot of the distribution of the $pCO_2$ in the surface waters of the Aegean Sea revealed that the area is causing a net $CO_2$ uptake from the atmosphere (Krasakopoulou et al., 2009a).
A "spring and late summer-early autumn" snapshot (in 2008) showed that the Aegean acted as a sink for atmospheric $CO_2$ during spring, and as source of $CO_2$ during the warm period (Krasakopoulou et al., 2009b). Recently, González-Dávila et al. (2016) based on one-year pH records in Outer Saronikos Gulf, reported that the area was acting as a source of $CO_2$. The few recent in situ measurements of $A_T$ in the Cretan Sea suggest that the existing $A_T$-S relationships for the Mediterranean do not reproduce efficiently the $A_T$ levels usually observed in
the Aegean Sea (Krasakopoulou et al., 2015).

     The significant variability in the circulation patterns (Theocharis et al., 1999; Georgopoulos et al., 2000; Korres et al., 2014) and the strong coupling with the biochemical processes in the Cretan Sea (Tselepides and Polychronaki, 2000) has made evident that sparse spatial and temporal observations are prone to misrepresentation of the underlying dynamics.

**2.2 Oligotrophy, atmospheric deposition, primary production, plankton stock and biodiversity hot spots**

     The Eastern Mediterranean is characterized by an anomalous nitrogen to phosphorus ratio ranging from 25 to 28, significantly higher than the normal oceanic Redfield ratio (16:1) (Kress and Herut, 2001; Kress et al., 2003). In this basin, riverine nutrient inputs are very low (Koçak et al., 2010), and atmospheric deposition is
believed to be the main source of nutrients in the euphotic zone of the open sea, other than the vertical mixing of water during winter (Christodoulaki et al., 2013 and references therein). Interactions between atmospheric deposition of nutrients and ocean productivity can be critical for both $CO_2$ storage in the ocean and marine ecosystems life. Moreover, the Mediterranean atmosphere is a cross road for air masses of distinct origin, highly affected by both natural and anthropogenic emissions into the atmosphere that strongly interact chemically, due
to the high photochemical activity in the area, leading to the formation of nutrients such as nitrogen compounds (Vrekoussis et al., 2006; Finlayson-Pitts, 2009). Dust aerosols from the African continent are also affecting the area as carriers of nutrients, such as phosphorus and iron (Gallisai et al., 2014). Although the impact of this input upon primary productivity and key biochemical processes has been studied (Kouvarakis et al., 2001; Krom et al., 2004, 2010; Okin et al., 2011; Lazzari et al., 2011; Christodoulaki et al., 2013), the exact contribution to the
balance of nutrients and the resulting impact of atmospheric deposition on the productivity of the oceans and climate remains uncertain and deserves further investigation (Duce et al., 2008).

     The Eastern Mediterranean Basin is considered to be an ultra-oligotrophic system, in terms of both primary productivity and Chl-a concentration, compared to the rest of the Mediterranean and to other oceans (Siokou-Frangou et al., 2010 and references therein). Based on a division of the Mediterranean in bioprovinces



(as derived from SeaWiFS Chl-a), the South-Central and Eastern Mediterranean (including the Cretan Sea) are clustered together within 60% of the total Mediterranean area. These areas are characterized by pronounced interannual variability, with generally oligotrophic conditions interrupted by pulses of biomass enhancement (D'Ortenzio and Ribera d'Alcala, 2009). The strong oligotrophy of the Cretan Sea has also been witnessed by

primary production investigations (Ignatiades, 1998; Gotsis-Skretas et al., 1999; Psarra et al., 2000) giving an annual primary production of 24.8 gC m$^{-2}$ (Ignatiades et al., 2002). Oligotrophy in the Cretan Sea has been also suggested by the higher relative abundance of nanociliates (Pitta and Giannakourou, 2000), heterotrophic bacteria and nanoflagellates as well as by the level of bacterial production (Christaki et al., 1999; Van Wambeke et al., 2000). Mesozooplankton abundance in the Cretan Sea's epipelagic layer was found to be at the same level as in

the open Ionian and Levantine and community composition have shown significant similarities with the above areas (Siokou-Frangou et al., 2010 and references therein). The dominance of the multivorous food web (herbivorous and microbial grazing modes having similar roles) in the Cretan Sea (Siokou-Frangou et al., 2002), is a dissimilarity with most areas of the Mediterranean Sea where the microbial food web generally dominates (Siokou-Frangou et al., 2010 and references therein).

The Mediterranean Sea also constitutes a hot spot of biodiversity with a uniquely high percentage of endemic species (Coll et al., 2010 and references therein; Siokou et al., 2010). However, biodiversity studies are still limited in the Mediterranean both for benthos (Danovaro et al., 2010) and plankton, with the studies of the latter being mainly limited to the epipelagic layer (Siokou et al., 2010). Marine diversity estimates are especially limited for deep-seas and portions of the southern and eastern regions, with microbes' diversity being substantially

underestimated (Danovaro et al., 2010). In addition, the Eastern Mediterranean is more subject to change by the invasion of alien species in combination with warming (Coll et al., 2010), due to the connection with the Red Sea. All the above make clear the interest of studying the biodiversity of the open Cretan Sea.

### 2.3 Biological pump efficiency at mid and deep waters

The biological carbon pump is the major oceanic process that photosynthetically converts the dissolved

CO$_2$ in the surface layers of the ocean to particulate organic carbon, which is then consumed by pelagic biota, exported to depth by a combination of sinking particles, advection or vertical mixing of dissolved organic matter, and transport by animals, and/or sequestered in the deep sea. Buesseler and Boyd (2009) have noted that ''the surface ocean'' is ''where the 'strength' of the biological pump is set'' whereas ''the subsurface ocean'' is ''where the 'efficiency' of the biological pump is determined.'' The magnitude and efficiency of the biological carbon

pump is controlled by physical and biogeochemical processes in surface, mesopelagic and bathypelagic layers, varying at daily, weekly, seasonal and inter-annual timescales. Several sediment trap experiments have been carried out over the past decades in the Mediterranean, both in continental shelf and open-sea environments, studying mass fluxes, budgets of particulate organic carbon and other major and minor elements, and variability of particle fluxes and mechanisms controlling the transfer of particulate matter to the open sea (Stavrakakis et al.,

2013 and references therein). Two of the most recent studies that have been conducted in the Eastern Mediterranean, revealed that only a small portion of primary production (0.3-0.5%) reaches the bathypelagic zone (Gogou et al., 2014; Stavrakakis et al., 2013).

The inhabitants of the deep sea, play an important role in determining the depths to which carbon is exported, a role mainly played by microbes and zooplankton (review by Turner, 2015). The lack of data from



midwater depths severely limits our ability to quantify the efficiency of the biological pump (Robinson et al., 2010). Although in the western Mediterranean there have been several studies on deep living zooplankton, much less is known for the Eastern Mediterranean (review by Saiz et al., 2014), although its stock and composition has been investigated (Siokou et al., 1997, 2013) and its importance in consuming sinking particles reported

(Koppelman et al., 2004).

In the Cretan Sea, deep water mesozooplankton has occasionally been studied (Siokou et al., 2013) and the vertical flux of zooplankton faecal pellets quantified (Wassman et al., 2000). In this area, macrozooplankton vertical migration appears to occur at diel and seasonal scale down to 500 m as suggested by ADCP measurements (Cardin et al., 2003; Potiris et al., this issue). Since zooplankton vertical migration may constitute an important

active vertical flux increasing the biological pump's efficiency (review by Frangoulis et al., 2005), the role of this vertical migration in the Cretan Sea pump needs exploration.

### 2.4   Basin to global anthropogenic impact

Although it is clear that human activities have modified the biogeochemical cycles of nutrients in terrestrial and aquatic ecosystems (Galloway et al., 2004, 2008), the understanding of the marine ecosystems'

response to atmospheric and terrestrial input variations is limited. This applies especially to oligotrophic areas, where nutrients are the dominant ocean productivity limiting factors (Powley et al., 2014; Van Cappellen et al., 2014). An increase in the N:P ratio in the ocean seems to have been driven by an increase in the atmospheric deposition of nitrogen due to human activities (Christodoulaki et al., 2016), although, potential changes in the atmospheric phosphorus deposition may also have contributed to this. Anthropogenic atmospheric inputs of

phosphates to the ocean are 15% globally. In oligotrophic oceanic areas, such as Eastern Mediterranean Sea, where productivity may be limited by phosphorus, these inputs may contribute as much as 50% to the deposition (review by Mahowald et al., 2008).

Besides atmospheric deposition, a large part of the Mediterranean coasts host urban and/or industrial areas with coastal water contamination occurring by pollutants, especially in the vicinity of large cities, with direct

impact on marine ecosystems (EEA, 2015). In addition, changes on the quality and quantity of river inputs have an impact on marine ecosystems, although this is evident only in the productive coastal areas (Ludwig et al., 2009). Furthermore, vertical mixing in the Mediterranean Sea is an important mechanism for the fertilization of the surface waters during the mixing period of the water column. Under global warming, a higher surface temperature would result in a stronger stratification of the water column (Barale et al., 2008) thus increasing the

importance of the external sources of nutrients like atmospheric deposition. However, thermal instability in the atmosphere, occasionally induced by air pollution, could lead to strong convective events in the atmosphere that could also increase the vertical mixing in the water column, bringing limiting nutrients like P to support the phytoplankton growth (Christodoulaki et al., 2016).

The Cretan Sea, based on a delineation of the Mediterranean basins according to the river discharges, is

qualified as having open ocean characteristics (Ludwig et al., 2009 and references therein). However, river discharges only do not allow to fully support the assumption that this area receives only basin to global scale anthropogenic impact. In fact, it has still to be verified against other poorly known direct open ocean anthropogenic activities in this area, such as fisheries (including deep ecosystems exploitation), maritime transport and noise.





**3    Aims and mission**

POSEIDON (www.poseidon.hcmr.gr) is an observatory research infrastructure of the Eastern Mediterranean basin, for the monitoring and forecasting of the marine environment, supporting the efforts of the international and local community and replying to the needs and gaps of science, technology and society (Perivoliotis et al.,
2018). It was developed in three phases under the funding of EEA Financial Mechanism (85%) and Greek National funds (15%): POSEIDON-I (1997-2000), a first-generation buoy monitoring network with operational centres, forecasting system, and relevant human resources; POSEIDON-II (2005-2008) a system upgrade and expansion; and finally POSEIDON-III (2009-2011) a deep sea observing capacity expansion. Recently (2017), an extended upgrade and renewal of POSEIDON buoy network monitoring parts and components and their supporting
hardware was realized due to the implementation of an integrated marine monitoring program funded by the EEA Financial Mechanism 2009-2014. The POSEIDON's biochemistry-ecosystem observational component aims are presented below.

A. The **Management aims** based on the experience acquired by POSEIDON team and international management recommendations (NRC, 2011; Karl, 2010; Ruhl et al., 2011) are:

a.   Sustainability through prioritization and cost-effective management. This aim guides the expansion of the number and of the spatiotemporal coverage of the parameters studied (see Sect. 7)

   b.   Integration of disciplines, platforms and analysis methods, by bridging various measurement scales and methodologies as well as through a step-by-step integration of an increasing number of different biochemical platforms (see Sect. 4), parameters and analysis methods (see Sect. 5)

c.   Maintain a quality controlled biochemical-ecosystem database, with reduced data delivery time and expanded accessibility (see Sect. 6)

   d.   Support ecosystem model validation for improved ecosystem forecasting and management

   e.   Provide a technology test bed for new biogeochemical sensors

   f.   Maintain collaboration through strong networking with similar observatories for common strategy,
complementary tasks, protocol standardisation, and transfer of knowledge (see Sect. 4)

B. The **scientific objectives** coming out from the international (Ruhl, 2001; IMBER, 2005, 2010; NRC, 2011), European (EGMRI, 2013) as well as national scientific projects, are to provide scientific knowledge and support on the study of:

   a. ocean mechanisms, including their interactions and spatiotemporal variability (duration,
occurrence, match-mismatches). The currently examined mechanisms are: solubility and biological pump, transformation and transfer of matter (fluxes within food webs and in-out of ocean interfaces), food web structure (stock, functional groups, microbial loop, size spectra) and ocean-atmosphere interactions (heat, dust, $CO_2$).

   b. biodiversity unknowns (number of species present and their temporal variability), in order to help
understand species interaction with food web functioning and biogeochemical cycling.

   c. the sensitivity of biodiversity and the variability of ocean mechanisms in relation to combined natural forcing factors and anthropogenic pressures. The current pressures considered, are those induced by warming (temperature, circulation, stratification, wind mixing, extreme-episodic events), ocean acidification (carbonate chemistry), nutrient dynamics (stoichiometry, dust deposition) and



oxygen concentration. Other pressures and feedback effects planned to be considered in the future studies of the observatory are presented in the Sect. 8.

C.  Provide services to marine **policy-makers and the society,** while adaptation to the evolving needs is ensured by a feedback mechanism put in place. POSEIDON is being developed in accordance to the policy frameworks suggested by IOC/GOOS, EuroGOOS, MonGOOS and GEO. The achievement of these objectives is made through collaboration with scientists from other disciplines including social scientists, and in communication with policy makers and the wider public. Within this framework the observatory aims to:

    a.  Provide policy makers with sound scientific knowledge to assist in making informed decisions. The observatory provides several MSFD descriptors and tests indicators of ecosystem health status.

    b.  Promote understanding of anthropogenic impact on the ocean systems, and at the same time the dependence on them.

    c.  Provide products to the end user through the POSEIDON operational biochemistry and ecosystem modelling tools.

A balance between the operational and research character of the infrastructure is maintained through the integration of methodologies and tools developed in relevant EU initiatives and projects. In parallel, the objectives are adapted based on new science and society goals and future scopes are considered in the strategic plan (see Sect. 8).

## 4    Components-Platforms

The present (2017) status of the Cretan Sea's coastal-open ocean biogeochemistry, ecosystem and biodiversity observatory includes: a) an open sea fixed platform with a multi-sensor array, b) a coastal fixed platform with a multi-sensor array, c) open sea sediment traps, d) an open sea ADCP e) open sea sampling through regular R/V visits, f) coastal sampling through regular R/V visits and g) a Ferrybox (FB) (Fig. 2). It is complemented by land-based HCMR facilities (laboratories, mesocosms, calibration room) located on the island of Crete. The Cretan Sea observatory platforms are integrated within a wider area network of Eulerian and Langragian (Bio-Argo floats) BGC platforms. Remote sensing (satellite) and BGC-ecosystem modelling products are used for the validation of measurements and vice versa. Last, but not least, comes the experienced multitasking personnel with significant experience in all aspects of the system.

We describe below these components via an historical evolution describing the progressive introduction of several platforms operating in long-term, and their development towards a more biogeochemical-ecosystem orientation.

### 4.1   Fixed platforms. From physics to biogeochemistry

In the framework of EuroGOOS, a multi-national effort to develop an integrated operational monitoring and forecasting system for the Mediterranean Sea took place under the Mediterranean Forecasting System (MFS) project (Pinardi and Flemming, 1998). During the Pilot Phase of the Project (1998-2001), a significant element of the designed observing systems was the Mediterranean Moored Multi-sensor Array (M3A) deployed in the open Cretan Sea, a prototype observatory that was designed to form the basis of a permanent network of moored stations



for continuous recording of open-ocean conditions in the Mediterranean Sea (Nittis et al., 2003). From the eight planned stations in the Mediterranean, the E1-M3A was the prototype, the first open ocean buoy deployed in the Mediterranean in January 2000, followed by two other M3A stations in the Adriatic and Ligurian Sea few years later.

The nowadays called POSEIDON E1-M3A buoy (WMO 61277) is the founder component of the Cretan Sea observatory. The current configuration was designed and built during the second phase of POSEIDON. The mooring is located about 24 nautical miles north of the island of Crete at a depth of 1400 m (Fig. 1), and is currently the most developed physical-biogeochemical observing site of the POSEIDON system (Petihakis et al., 2006). Biochemical sensors currently deployed on the buoy are $O_2$, fluorescence (Chl-a) and pH (described in
detail in Sect. 5). It is considered a reference point for monitoring open ocean biochemical processes (including air-sea interactions) of the Eastern Mediterranean and part of the operational oceanography observing system developments supporting the MSFD implementation in the Mediterranean Sea (Pinardi et al., 2009). Consolidating on the long experience of physical parameter monitoring, the objective of the observatory has been recently expanded to include regular monitoring of the epipelagic ecosystem and the associated biochemistry (see Sect.
15   4.2).

Recently (2016), the Cretan Sea observatory was expanded with a second fixed platform, the Heraklion Coastal Buoy (HCB) (Fig. 1). This Oceanor Seawatch buoy, coupled with the existing infrastructures (open sea buoy, Ferrybox, R/V missions, Bio-Argos) in the area, opens possibilities to study the interaction of coastal with open sea processes (e.g. wave dynamics, exchanges of matter, extreme events spatial extend).

Outside the Cretan Sea, but within the POSEIDON network two, similar to E1-M3A, Oceanor Wavescan buoys were deployed in the N. Aegean and Ionian waters. Both these buoys provide meteorological, physical, and since 2017, also biochemical ($O_2$, Chl-a) data. The buoy placed between Athos peninsula and the island of Lemnos (AB) (Fig. 1), monitors an area affected by the Black Sea water entering the North Aegean through the Dardanelles straits, which plays a significant role modulating the thermohaline characteristics and dynamics of the whole
Aegean Sea. The Pylos site (PB) in the SE Ionian is a crossroad where intermediate and deep-water masses meet. The site is located on the pathway of the Aegean Sea dense water that travels to the north along the western coast of Greece.

### 4.2   Water column sampling and sediment traps (RV cruises). From occasional high spatial resolution snapshots to sustained temporal coverage

The biochemistry of the Cretan Sea has been studied approximately every 2 to 5 years since the 80's by in situ sampling using R/V's in the framework of different research programs (e.g. POEM, PELAGOS, CINCS, MATER, SESAME, PELAGIAL). However, having a wide spatial coverage, most of the cruises were at the best made at seasonal scale (generally at mid-spring and early-autumn). In addition, there was a large sampling gap between 1999 and 2005 (Fig. 3) while only from 1993 to 1995, during the CINCS project, high frequency (bi-monthly)
samplings were performed (Danovaro et al., 2000; Psarra et al., 2000; Tselepides and Polychronaki, 2000; Van Wambeke et al., 2000). Parallel to this, sediment traps were also deployed in the Cretan Sea during the CINCS project 1994-1995 (Stavrakakis et al., 2000) and MATTER project 1997-1998 (Lykousis et al., 2002).

A major contribution towards the implementation of a long term biogeochemical monitoring based on regular R/V visits was made in 2010, when the Institute of Oceanography of HCMR initiated a continuous




sampling program at the POSEIDON E1-M3A site (Fig. 3). Initially it included conductivity-temperature-depth (CTD) casts and seawater/plankton sampling providing multiple parameters (e.g. Chl-a shown in Fig. 4). Latter on (2011) it was complemented with two sediment traps, and a year later (2012) with an acoustic Doppler current profiler (ADCP) (see Sect. 4.3). Since 2016 CTD casts and seawater/plankton sampling are also performed at
monthly frequency next to the coastal fixed platform HCB. These initiatives are maintained until present (2018).

### 4.3  ADCP. From currents to zooplankton vertical migration

A fixed position ADCP was first deployed next to the E1-M3A buoy location in 2000 for a period of one year. This 75 kHz ADCP placed at 600 m looking upward, allowed to study currents at multiple depths and gave indication of the presence of vertical migration of scatterers, probably large zooplankton (Cardin et al., 2003).
Acknowledging the importance of this particular mechanism in the modulation of the ecosystem, in 2012 the ADCP was redeployed (depth ~500 m), and it provided data to analyse the vertical migration patterns (at diel and seasonal scale) of large zooplankton from 400 m to the surface (Fig. 5; Potiris et al., this issue).

### 4.4  Ferrybox. From temperature and salinity to carbonate system

A Ferrybox system operated for the first time in the Mediterranean, on the route connecting the ports of
Piraeus (Athens) and Heraklion (Fig. 1), for one year between 2003 and 2004 within the framework of the EU-funded project Ferry-Box (Nittis et al., 2006). This fully automated, flow-through system included sensors for underway measurement of temperature, salinity, fluorescence and turbidity. High frequency measurements took place every night, and the data were delivered in Near Real Time (NRT). The system was reactivated in 2012 and operates since then (with a gap from late 2014 to mid 2017), with an upgrade of the components, that now include
$O_2$, $CO_2$ and pH sensors. The FB has been proven a helpful tool in the study of water circulation (e.g. modified Black Sea Water flowing in the Aegean Sea), in particular when assimilated into prognostic numerical circulation models to improve their accuracy (Korres et al., 2014). For surface Chl-a validation/calibration, a comparison of FB's fluorescence (e.g. Fig. 6) with satellite data as well as nearsurface measurements of Chl-a (by discrete water sampling) from nearby R/V cruises is a powerful multi-tool used by the observatory scientific team.

### 4.5  Personnel. From separate specialists to multitask collaborators

Specialized personnel is the major component for a smooth, continuous functioning of any multiplatform-multidisciplinary observatory. Since its beginning, POSEIDON invested in the necessary human resources, namely a dedicated group of scientists and engineers that operate, maintain and upgrade the system on a full-time basis. Over the last years as the number and complexity of platforms significantly grow and as the inflow of data
increased, the knowledge level of personnel became more demanding. Balancing these demands with personnel shortage, due to budgetary constraints, led not only technicians but also scientists to have multifunctional duties that "...*may involve aspects of a bosun, chemical safety officer, satellite communications specialist, network administrator and electronics technicians*" (NRC, 2011). In addition, despite the need of specialists of each domain remained, it appeared necessary to share the basics of each domain among several persons (e.g. 3 having
knowledge of buoy maintenance-deployment, 5 of seawater sampling, 3 of CTD cast data processing, etc). Through active communication and with the support of various EU projects (JERICO, FixO3, JERICO-NEXT,





EMSODEV, EMSOLINK) the personnel acquired precious experience by strong collaboration, continuous education and exchange of knowledge-experience with personnel of similar observatories.

### 4.6 BioArgo floats

During 2012 the Greek Argo infrastructure was launched as a component of the POSEIDON observing system, with the aim to purchase and deploy 25 ARGO floats (www.greekargo.gr), further contributing to the international ARGO community efforts to monitor the Eastern Mediterranean region. In 2016, five of the 15 Greek deployed floats were Bio-Argos (equipped with $O_2$ sensor). It is worth noticing that in the Aegean Sea the ARGO recordings are largely based on POSEIDON floats.

### 4.7 Gliders

Two SeaExplorer gliders were recently (2017) added to the monitoring platforms of the POSEIDON system. The two gliders will be gradually integrated to the operational network of the system with the ultimate objective of establishing at least two endurance lines in the Aegean and Ionian Seas. In the Cretan Sea, the continuous monitoring through an endurance line is expected to contribute to the further knowledge of the seasonal variability of the flow field, collecting also evidences of the intermediate or deep-water formation events that are known to occur in the area.

### 4.8 Biogeochemical Modelling

Forecasting tools are centrally placed in the POSEIDON system, with a number of state-of-the-art weather, wind waves, ocean circulation and marine ecosystem numerical models, initialization and data assimilation schemes providing 5-days ahead information on daily basis regarding the atmospheric (Papadopoulos et. al., 2002), sea state (Korres et., 2011) and hydrodynamic conditions (Korres et al., 2010) in the Aegean/Ionian Seas and in the Mediterranean. Currently, the POSEIDON modelling group is providing the wave forecasting products of the Copernicus Marine Environment Monitoring Service (CMEMS) for the Mediterranean Sea in the framework of MED-MFC.

The POSEIDON ecosystem simulation tool is one of the first developed in the Mediterranean, producing daily forecasts for a range of ecosystem parameters for the whole basin. The Cretan Sea has been a test site for the implementation of the biogeochemical European Regional Seas Ecosystem Model (ERSEM, Baretta et al., 1995), since it is the only offshore site in Greece in which the ecosystem has been systematically observed, providing a very successful test bed for model development (Petihakis et al., 2002; Triantafyllou et al., 2003a, b, c; Hoteit et al., 2004; Christodoulaki et al., 2013; Kalaroni et al., 2016). General calibration – validation activities are applied to the operational models, as data from the observatory are used in conjunction with experiments (e.g. mesocosms, see Sect. 4.9), for the analysis and modelling of specific processes, such as the microbial functioning, the effect of the atmospheric deposition (Christodoulaki et al., 2013; Tsiaras et al., 2017) or for the assimilation algorithms of sea colour data in BGC models (Kalaroni et al., 2016).



### 4.9 Land based support facilities

Sensor maintenance and analysis of discrete samples are not the only reasons for having several land-based facilities located nearby the observatory. Other land-based infrastructures such as calibration room, micro- and mesocosms, meteorological stations, and atmospheric deposition station, provide added value to the observatory.

The Poseidon calibration lab (based in Heraklion) accommodates the regularly scheduled calibration of sensors, considering the local environmental conditions (e.g. for conductivity sensors high salinity and for Chl-a sensors low concentration and native phytoplankton species). The lab is equipped with a specially designed large calibration tank and several smaller fiberglass tanks. Several reference sensors and equipment allow calibration of temperature, conductivity, fluorescence (Chl-a), turbidity and dissolved oxygen sensors. It has been proved a

powerful tool for the calibration of sensors deployed in the wider Mediterranean Sea (Pensieri et al., 2016). In particular, calibrations of fluorometers and turbidity sensors are especially important in the Mediterranean, and more so in the Eastern part, since primary production is significantly lower compared to most other areas in the world where such sensors are operating (Bozzano et al., 2013).

Microcosm laboratory facilities allow performing experiments on the physiology of a specific species or

group of organisms of the Cretan Sea. In addition, the land-based mesocosm facilities at Crete (www.cretacosmos.eu), part of the MESOAQUA network of European mesocosm facilities, allow to better explore specific biogeochemical processes in connection with food-web functioning occurring within the Cretan Sea oligotrophic ecosystem.

Finally, the "Finokalia" atmospheric deposition monitoring site of the University of Crete located on the

island of Crete (www.finokalia.chemistry.uoc.gr) is among the few atmospheric stations located along the coasts of the Eastern Mediterranean, with the particularity of being representative background station for atmospheric observations in the area (Kanakidou et al., 2011). The proximity of Finokalia monitoring station with the Cretan Sea observatory offers a unique coupling of observing locations to study the impact of atmospheric deposition in the Mediterranean Sea (e.g. Kouvarakis et al., 2001).

### 4.10 Connection with European and international observatories

A regional observatory can reach a higher potential when it belongs to a wide network of observatories (Ruhl et al., 2011). Ocean observation at global scale is implemented through the international program GOOS (Global Ocean Observing System) executed by the Intergovernmental Oceanographic Commission (IOC) under the auspices of UNESCO (www.ioc-goos.org/), aiming to inform scientists, policymakers and society (Karl, 2010).

GOOS is a global network of ships, buoys (fixed and drifting), subsurface floats, tide gauges and satellites that collect real time data on the physical state as well as the biogeochemical profile of the world's oceans for three critical themes: climate, ocean health, and real-time services.

At the European level, the EMB and EuroGOOS (www.eurogoos.org) have joined forces towards a truly integrated and sustained European Ocean Observing System (EOOS) as suggested by the Ostend Declaration

(EurOCEAN 2010 Conference) and described in the EMB position paper "Navigating the Future IV" (EMB 2013).

For the open ocean, the OceanSITES project, an integral part of GOOS, facilitates international coordination of time-series at fixed locations (www.oceansites.org). The E1-M3A buoy of the Cretan Sea observatory is part



of the global network OceanSITES, via its European contribution (EuroSITES and former EMSO ERIC), which established integration between the fixed-point deep ocean observing systems around Europe. A strong and direct collaboration is established with scientists from observatories operating in the Mediterranean focusing on open ocean biochemistry (DYFAMED, W1-M3A and E2-M3A) as well on the atmospheric deposition monitoring

(Finokalia). The open-ocean observatory of the Cretan Sea has been expanded with a multi-platform coastal component (including coastal buoy and Ferrybox) in the framework of JERICO-NEXT project in an attempt to characterize the timing, intensity, and fate of organic matter and the coupling of offshore with coastal processes.

## 5   Biogeochemical, ecosystem parameters

The complexity of ocean environment has led the international scientific community (e.g. GOOS Expert
Panels) to identify essential ocean variables based on their relevance, feasibility and cost effectiveness (www.goosocean.org). For comparison and consistency, it is also crucial for local observing systems to be an integral part of a wider network of observatories. The choice of variables and protocols used at the Cretan Sea observatory are aiming to follow international relevance, best practices, inter-comparable methodologies and are evolving towards more simplified and cost-effective sensing/sampling/analysis. The Cretan Sea observatory's
biogeochemical parameters, associated platforms and their spatiotemporal coverage are summarized in Fig. 7, and detailed in the Sect. 5.1 and 5.2.

### 5.1   Biochemical Sensors

Table 1 presents the BGC and associated parameters measured by sensors on the various platforms of POSEIDON. The operation protocol for these sensors follows international best practices for standardized
methods, regarding checking and maintaining sensor accuracy, on demand manufacturer calibration, regular lab calibrations (Coppola et al., 2016) and comparisons with relevant in situ data. Pre-deployment lab calibrations for dissolved oxygen (DO), turbidity and Chl-a (fluorescence) sensors are performed in the POSEIDON calibration lab (described in Sect. 4.9). In situ comparisons ("field calibrations") are made for DO, pH and Chl-a (fluorescence) sensors, by comparison with casts made with a reference CTD (regularly calibrated) and with
discrete samples taken at the depth of deployment of the sensors.

### 5.2   Biochemical sampling and analytical (lab) methodology

Seawater and plankton samples are taken regularly in the vicinity of the fixed biochemical platforms and on-board the FB. In addition, samples of sinking particulate matter are collected with sediment traps. The parameters obtained and the lab methods used for analysing these samples are summarized in tables 2 and 3.
Water-plankton sampling is made using either a) a large (62m) or medium (23m) R/V, or b) a small R/V (RIB boat). The small R/V offers the opportunity of maintaining long time series at a lower cost, is easier to schedule, and is twice as fast as the larger vessels (allowing thus quick storage and analysis of samples at the land facilities). Overall, experience acquired up to now showed the small R/V to be the most frequently used vessel. HCMR has therefore invested to grow up its capabilities (e.g. new winch allowing casts much deeper than 1000
m – Fig. 8).



During the R/V visits in the area of the fixed biochemical platforms the order of sampling is always CTD-Niskin bottle casts followed by net tows. The vast majority of casts and net tows (>80% after 2010) were made between 09h00 and 14h00 (local time). The order of sampling, the sampling procedure and storing protocols follow the international recommendations described by Lorenzoni and Benway (2013) and remain consistent since

2010, with recording of cruise metadata (Cruise Summary Reports available at www.seadatanet.org). Adaptations of certain protocols have been made in certain cases, considering the open ocean oligotrophic conditions of the area. For example, zooplankton sampling was initially made with a single 200 μm mesh size net but soon it was changed into parallel two mesh-size nets (45 μm mesh size and 200 μm mesh size) tows. This was done due to the small size organisms found in the area, since combined data from several nets allows for a better estimation

of the mesozooplankton biomass (Frangoulis et al., 2016). In all cases, when a new method was introduced, a minimum one-year period of sampling-analysis using both old and new methods was applied, before moving exclusively to the new method (e.g. Chl-a by fluorescence and by HPLC where measured in parallel during 3 consecutive years).

### 5.3    Derived biochemical-ecosystem parameters and model state variables

Any necessary unit conversion of physical and chemical oceanographic data for model validation, is generally straightforward. However, many biological parameters expressed in biomass units are usually derived from the initial data with some conversion factor assumed. These conversions introduce biases that have to be carefully considered by biologists and modellers (Flynn, 2005; Frangoulis et al., 2010; Tsiaras et al., 2017). In the case of the Cretan Sea biochemical-ecosystem model, the choice of appropriate conversion factors from literature,

that can be considered, is made by the data providers (biologists) in communication with modellers and is revised regularly.

Besides units' conversion, a provider-modeller communication is mandatory as the numerous measured biological parameters often have no direct correspondence with the state variables of the operational biogeochemical model. Fig. 9 summarizes the correspondence between measured parameters and ERSEM model

variables used by the POSEIDON team for the Cretan Sea. Most plankton groups measured (except bacteria and diatoms) do not have a direct correspondence with a model variable. To make such links several plankton groups have to be grouped together, whereas others have to be split into subgroups based on their size and/or trophic functioning (e.g. Tsiaras et al., 2017). Planktonologists and the modellers of the observatory agreed that there is no rule of thumb, and model-field data correspondences should be adjusted according to the season and region.

### 30    6    Metadata and data handling

The data collected from the different platforms undergo Quality Control procedures according to EuroGOOS and OceanSITES working group standards and methodologies. The POSEIDON database is set to include BGC and BGC-associated parameters, either remotely sensed (pH, pCO$_2$, Chl-a, O$_2$, meteorological, T, S), or obtained from in-situ sampling (Chl-a, O$_2$, nutrients, plankton stock). The POSEIDON Data Centre, as the regional data

collection unit of the CMEMS, provides the compatibility of the recorded data with the existing large European data infrastructures (EMODnet, CMEMS and SeaDataNet). Data can be visualized through the POSEIDON web-site (fixed platforms, Ferrybox) and the MonGOOS data portal (http://www.mongoos.eu/data-center, all platforms





except sediment traps, ADCP), while the data are freely available to the public, the stakeholders and the scientific community, acknowledging the EC INSPIRE directive, in order to enable easy access to data and their reuse.

## 7   Sustainable development

A critical issue for any observatory is its sustainability. A continuous funding which will allow not only the day-to-day operations, but also the upgrade to the current state of the art technology is crucial. Unfortunately, in most cases marine observatories in Europe are developed through intermittent funding and national incentives. Likewise, the fixed platform of the Cretan Sea's observatory was founded through the FP6 EU Mediterranean Forecasting System Pilot Project (MFSPP) followed by the POSEIDON project and the EFTA funds. Furthermore, some activities and developments have been funded in the framework of both EU and national projects (research and infrastructural), while recently the observatory became part of the Hellenic Integrated Marine and Inland Observing Forecasting and Offshore Technology Systems Observing and Forecasting System (HIMIOFOTS), a research infrastructure that has been accepted as part of the Greek Research Infrastructure road map.

In periods when funds are limited, it is important to have and maintain a baseline via prioritisation of the parameters observed and platforms used. Such a plan must take into consideration, among other, the existing historical data sets, the international priorities and efforts and the specific scientific questions in the wider area (Eastern Mediterranean).

Despite some periods of low support, the multiparameter-multiplatform observatory approach, allows the participation to various targeted research projects and thus the provision of funds through multiple sources. In addition, the long experience in the entire chain of operations and the particular conditions of the Eastern Mediterranean make the observatory an excellent test bed both for new technology and sampling methods.

## 8   Future scopes, expansion and vision

The future of the observatory is presented in a short-term strategy and a long-term vision.

1. The short-term strategy of the biochemical observatory follows the expansion vision of POSEIDON system which considers recommendations, guidelines and priorities defined in the national Research Infrastructure road map of observing systems (HIMIOFOTS), review papers (e.g. Claustre et al., 2010), EU goals directives (MSFD, H2020), and visions of European (European Marine Board and EuroGOOS, e.g. EGMRI, 2013) and International coordinating bodies (GOOS, GCOS).

A main short-term goal is attaining a NRT character for the biogeochemical parameters together with a further expansion of the recorded parameters, with a greater focus in air-sea interaction. Based on the key parameters recommended by the EU (EGMRI, 2013), priority is currently given to further integrate sensors of $O_2$, $CO_2$, pH and fluorescence (Chl-a). Nutrient sensing is expected to be the next to follow, although the low concentrations found in the Eastern Mediterranean constitute a strong technological challenge. HCMR plans to expand the ability to host such biochemical sensors (with NRT capability) to more of the existing POSEIDON platforms (e.g. buoys, Bio-Argos, gliders, drifters, Ferrybox). In addition, a greater focus in the air-sea exchanges will be given. Finally, providing an operational status to the regular in situ sampling program is expected, following the experience gained and the standardisation of the procedures.



This strategy will give scientists the opportunity to study primary production, secondary (zooplankton) production, higher trophic level web structure, as well as feedback effects, such as the capacity to store $CO_2$, and the ecosystem's feedback on physics (light attenuation). In addition, the long-term time series data and the expanded NRT data delivery, proxy estimations, hazard mapping, and higher resolution predictions will benefit

numerous society users (local authorities, technical institutions, tourism industry, educational organizations, fisheries industry, environmental organizations, policy-makers etc).

2. The long-term vision considers further expanding the biochemical-ecosystems parameters and spatiotemporal coverage of the observational system, including the capacity to perform deep biochemical observations, that will allow to resolve mechanisms that remain poorly known, like benthic-water column

interactions, the functioning of mid-water and deep-water ecosystems, and the plankton vertical migration effect on active carbon flux.

Driven by other societal needs, supplementary underwater sensors will be considered. Once integrated in observing systems, these will offer scientists the potential to examine long term effects of additional pressures (e.g. thermal regime shift, contaminants including microplastics, noise, open ocean and deep ocean fishing,

harvesting) and additional products to society (e.g. contaminants warning systems).

**Acknowledgements**

We would like to thank M. Kouratoras for the infographics and all HCMR technical and scientific personnel that has contributed to discrete biochemical samples collection, analysis, and data processing at the POSEIDON E1-

M3A location. Part of the work has been funded by JERICO-NEXT project. This project has received funding from the European Union's Horizon 2020 research and innovation programme under grant agreement No 654410'

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





**Table 1: Current status of marine BGC and associated parameters (T, S) measured from sensors in the POSEIDON platforms**

| Parameter | Platform | Sensor |
|---|---|---|
| Temperature /Salinity | Buoys (E1-M3A, HCB, AB, PB) | SBE 16plus-IMP C-T-P<br>SBE 37 IM C-T |
| | Ferrybox | Thermo-Salinometer SBE 45 |
| | RV (CTD) | SBE 19+ OR SBE911 |
| Fluorescence (Chl-a) | Buoys (E1-M3A, AB, PB) | Wetlabs FLNTU |
| | Ferrybox | Scufa II Turner Design |
| | RV (CTD) | WET Labs ECO-AFL/FL 9 or Chelsea Aqua 3 |
| Dissolved Oxygen | Buoys (E1-M3A, AB, PB) | SBE43 /SBE 63/Aanderaa Optode |
| | Ferrybox | Aanderaa optode |
| | Bio-Argos | Aanderaa optode |
| | RV (CTD) | SBE 43 |
| Turbidity | RV (CTD) | WET Labs ECO FLNTU |
| PAR/Irradiance | RV (CTD) | Biospherical/Licor |
| pH | Buoy (E1-M3A) | Sensor LabpH |
| | Ferrybox | Meinsberg probe |
| | | |





**Table 2: Parameters measured from discrete bottle and net samples at high frequency (monthly). Method ranking from Lorenzoni and Benway (2013).**

| Parameter | Platform | Analytical method | Method Ranking |
|---|---|---|---|
| NO3+NO2, Si(OH)4 | E1-M3A, HCB | Manual Spectrophotometric | Acceptable |
| PO4 | E1-M3A, HCB | Magnesium-induced co-precipitation | Best |
| Total Chl-a | E1-M3A, HCB | Fluorescence and HPLC | Best |
| Other Phytopigments | E1-M3A, HCB | HPLC | Best |
| Viruses and Bacteria | E1-M3A | Flow cytometry | Best |
| Picophytoplankton | E1-M3A | Flow cytometry | Best |
| Nanophytoplankton | E1-M3A, HCB | Microscopy (UV + blue light excitation) | Best |
| Other nanoplankton | E1-M3A, HCB | Inverted microscopy | NA |
| Microphytoplankton | E1-M3A, HCB | Inverted microscopy | Best |
| Ciliates | E1-M3A | Inverted epifluorescence microscope (blue light excitation) | NA |
| Zooplankton | E1-M3A, HCB | 45μm and 200 μm nets, Scanning & Image analysis | Best |





**Table 3: Parameters measured from discrete bottle samples at low frequency (6 to 24 months) and from sediment traps (integrating 15 days). Method ranking from Lorenzoni and Benway (2013). +traps : measurement from both water column samples and from particulate matter in sediment traps. FB: Ferrybox.**

| Parameter | Platform | Analytical method | Method Ranking |
|---|---|---|---|
| Dis. Oxygen | E1-M3A, HCB | Winkler (against CTD sensor) | Best |
| $A_T$ | E1-M3A, FB | Potentiometric titration (Closed cell) | Best |
| DIC | E1-M3A, FB | Coulometric determination | Best |
| TEP | E1-M3A | Colorimetric determination | NA |
| POC/PN | E1-M3A (+ traps) | High Temperature Combustion via Elemental Analyzer | Good |
| DOC or TOC | E1-M3A | High Temperature Combustion | Best |
| TDN | E1-M3A | Persulfate Oxidation | Good |
| DOP | E1-M3A | Persulfate Oxidation | Best |
| Primary production | E1-M3A | 14C. Fractional day incubations scaled to daily rates | Acceptable |
| Bacterial production | E1-M3A | $^3$H-labelled leucine method | Best |





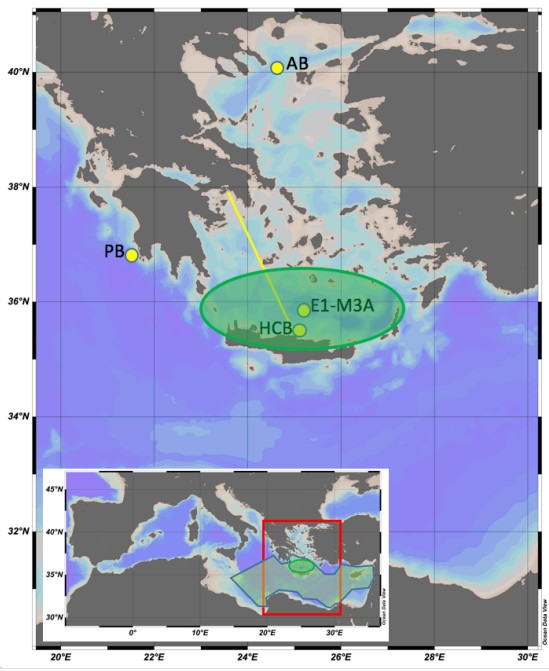

**Figure 1: Map of showing the location of the Cretan Sea (green ellipse) and all POSEIDON system BGC fixed platforms (yellow dots, see Sect. 4.1) and Ferrybox (yellow line, see Sect. 4.4). Inset map shows location within the Mediterranean Sea (red square) and E1-M3A spatial footprint (green area) for Chl-a using satellite observations (redrawn after Henson et al., 2016).**



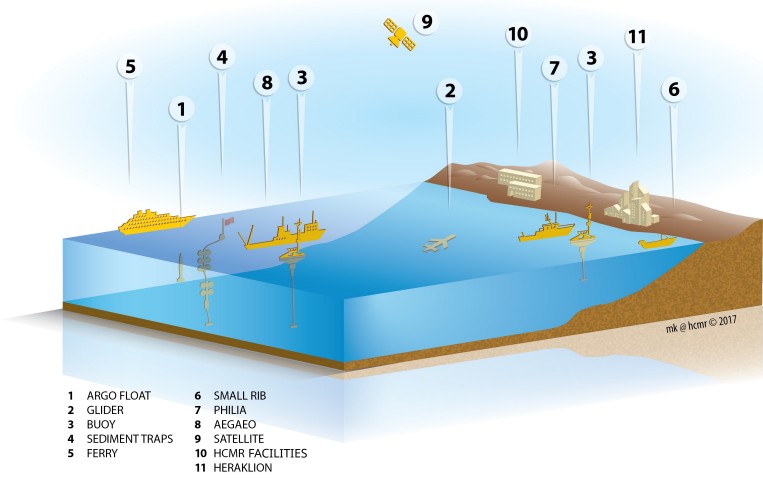

| 1 | ARGO FLOAT | 6 | SMALL RIB |
| 2 | GLIDER | 7 | PHILIA |
| 3 | BUOY | 8 | AEGAEO |
| 4 | SEDIMENT TRAPS | 9 | SATELLITE |
| 5 | FERRY | 10 | HCMR FACILITIES |
| | | 11 | HERAKLION |

**Figure 2: Platforms of the Cretan Sea biogeochemical-ecosystem observatory.**





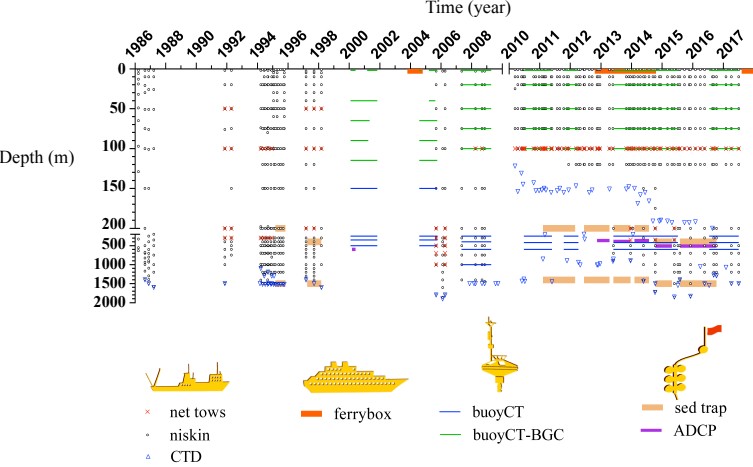

**Figure 3: Periods of operation of the different platforms located in the open Cretan Sea (historical metadata within a circle of 20 nautical miles radius around the position of E1-M3A; exclusion of metadata made within 10 nautical miles from a coast). Before 2010 the metadata listed may not be exhaustive. CTD casts and net tows were made from the surface to the depth shown in the figure. Ferrybox entry point is located at 3 m depth. BGC: $O_2$ and/or fluorescence sensors. CT: conductivity and temperature.**





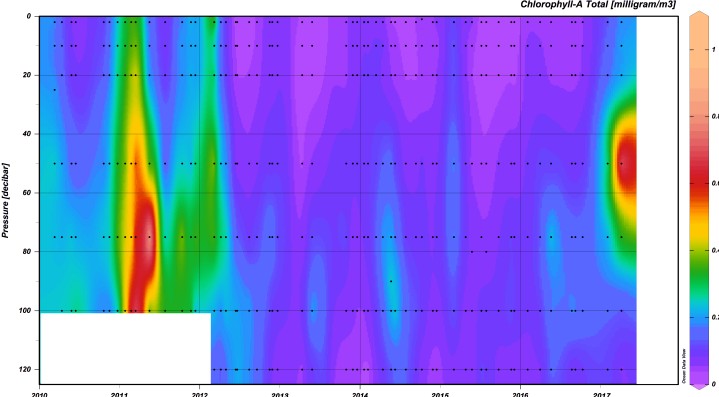

**Figure 4: Total Chl-a vertical distribution at the POSEIDON E1-M3A location from 2010 to 2017 (fluorometric analysis of seawater from bottle sampling).**





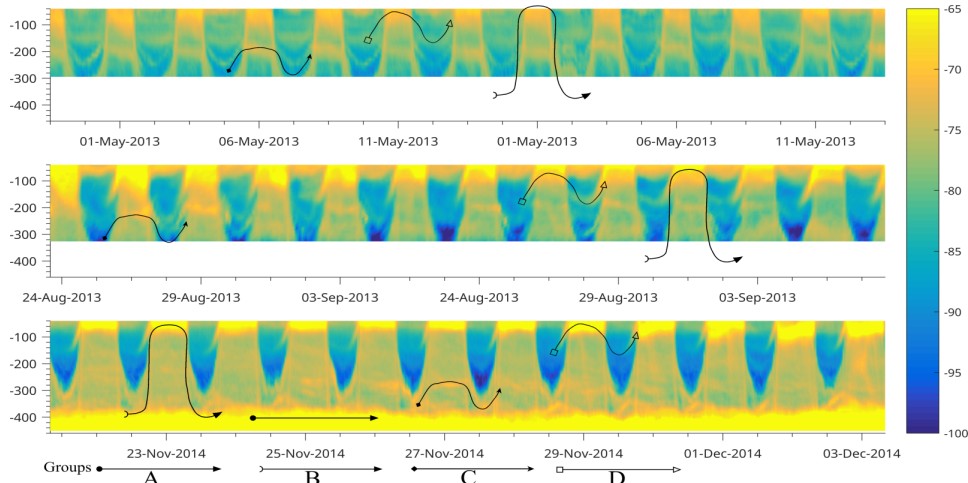

**Figure 5: Backscatter coefficient Sv from the 75 kHz ADCP placed at the POSEIDON E1-M3A location. Hand-drawn trails are attributed to different groups of zooplanktonic and micro-nectonic organisms (from Potiris et al., this issue).**





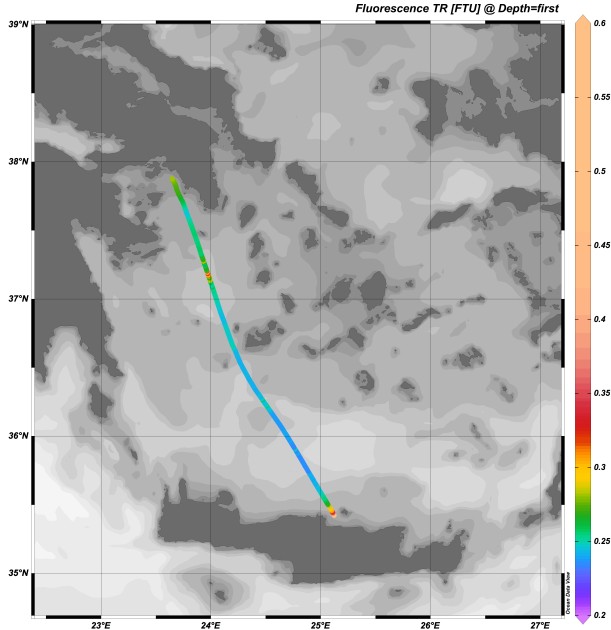

**Figure 6: POSEIDON Ferrybox's fluorescence recordings on 01/07/2012.**





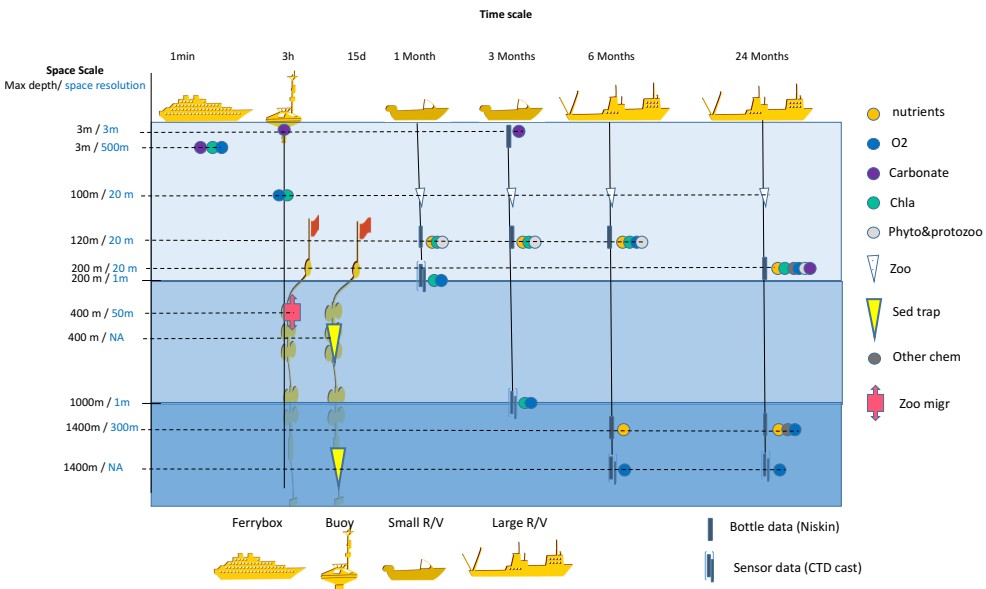

**Figure 7: Time and space resolution of biochemical data acquisition by the different platforms of the observatory. The list of parameters can be found in tables 1 to 3. Space resolution is vertical except in the case of Ferrybox. Carbonate: pH or $C_T$&$A_T$, Other chem: other chemical parameters, Sed trap: sediment trap, Phyto & protozoo: phytoplankton and protozoans; zoo: metazoans (collected with nets), Zoo migr: ADCP backscatter data for zooplankton migration.**





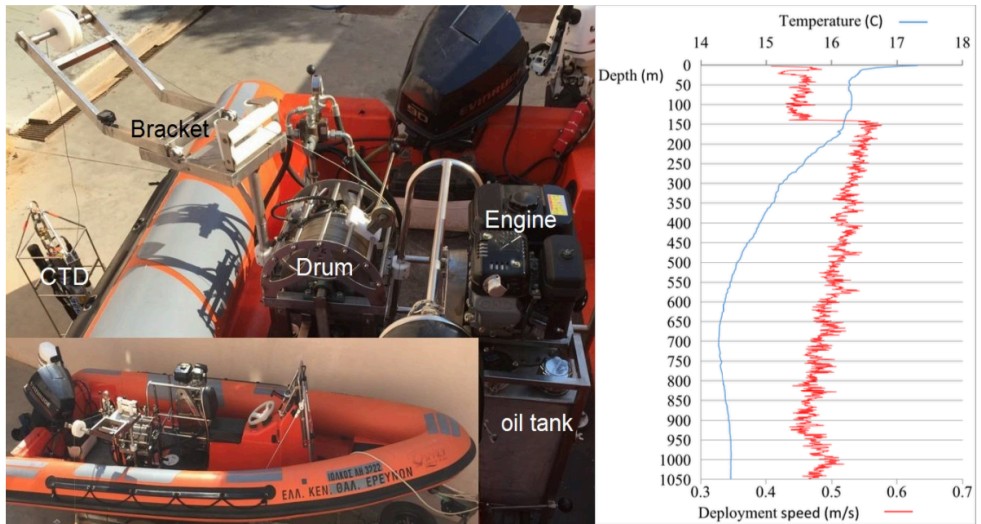

**Figure 8: Left: Hydraulic winch positioned on a small RIB allowing casts >1000 m (Pettas et al., 2015); Right: Temperature and deployment speed against depth from a CTD cast made using the hydraulic winch. Deployment speed decrease above 150m was made in order for the CTD to respond better to rapid environment changes like thermocline.**



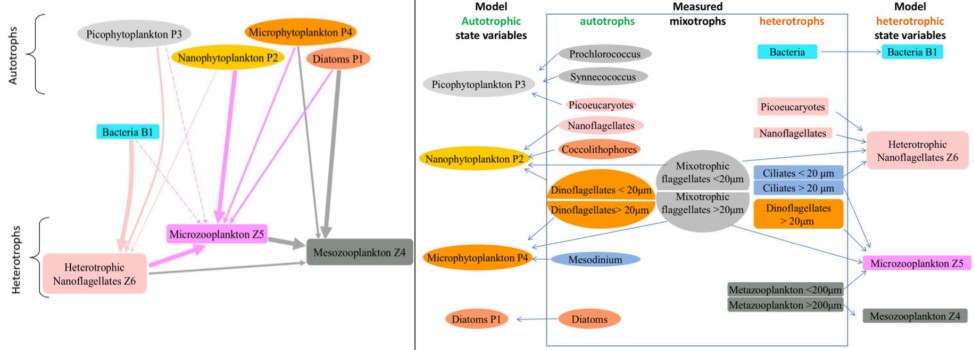

**Figure 9: ERSEM model's food web structure (left) and correspondence between model variables and measured parameters at the Cretan Sea observatory (right).**