# Peer review of "An integrated open-coastal biogeochemistry, ecosystem and biodiversity observatory of the Eastern Mediterranean. The Cretan Sea component of POSEIDON system."

_Ocean Science, 2018_

## Referee Comment (RC1) · Anonymous Referee #1 · 2 Mar 2018

**Journal: OS**
**Title: An integrated open-coastal biogeochemistry, ecosystem and biodiversity observatory of the Eastern Mediterranean. The Cretan Sea component of POSEIDON system.**
**Author(s): George Petihakis et al.**
**MS No.: os-2018-3**
**MS Type: Review article**
**Special Issue: Coastal marine infrastructure in support of monitoring, science, and policy strategies**

General comments

The present manuscript aims to describe the current POSEIDON observing system (OS) in the East Mediterranean Sea which is based on multi-platforms and multi-parameters approach. The actual status of this OS is composed by coastal and open sea eulerian and lagrangian platforms (gliders, Argo floats, Ferrybox, moorings and buoys). This OS delivers variables which are essential to understand and predict the impacts of climate change and anthropogenic pressure on marine ecosystems in the Eastern Mediterranean Sea.

The manuscript is well written but some parts are missing and some parts should be re-organized.

The section 2 is too long and it should be reduced. The authors should do a rapid state of arts of the situation in the eastern basin and describe the main characteristics of the basin (why is it important to implement and sustain an integrated OS in this region).

In the section 3, the scientific questions, which are the backbone of each OS, should be detailed in the first part followed by the management aims and services.

In the section 4, the personnel section should be moved after the section 4.9 otherwise this is confusing to see the personnel skills between the description of Ferrybox and Argo floats. In this section, there is no description of the POSEIDON buoys used for forecasting products. Why ?

After the data description in the section 5, a section on data analysis and data quality control is largely missing: How real-time data are managed ? Which protocols are used ? How the delayed mode data are adjusted ? Is there different correction levels (as for the remote sensing data) ? For example in Tables 1, 2 & 3, the authors should add some information on correction methods applied for each variables. In the same way, accuracy and measurements frequency should be added in these tables.

An additional section on the POSEIDON management and governance is missing too: who decide the choice of variables ? sensors ? how the new scientific needs are taken into account ? for example nowadays, global OS are focusing more and more on biological variables for marine ecosystems change (e.g. genomics). What is the vision of the POSEIDON group on this question ?

Regarding the scientific production, which is depending on the POSEIDON OS sustainability, how many publications, thesis, reports have been produced ? Any information regarding the DOI dataset statistics should be also included (at least on annex) to demonstrate the importance to maintain such OS.

Finally, a description and figure on integrated results from the different POSEIDON components are missing. This should prove the necessity to implement an integrated and multi-variables OS but also it will stress what are the actual gaps and needs.

Ancillary comments

The word "parameter" is often over-interpreted. A distinction with variable should be addressed here. Parameters are not natural quantities, variables are. A variable is an entity that changes over time or depth: this what we are measuring. In general, parameters are "constants" that define a specific instance of a general equation that is based on variables. That's why we used the term EOV for Essential Oceanic Variables.

"Biochemical" should be changed to "biogeochemical"

Figure 1
Information is missing regarding the glider endurance lines, the Argo floats observation area, the sediment traps deployment, etc… A new figure should be proposed with the names of seas, straits, main countries, …

Figure 3
This is not very clear and too small. It should be modified or removed

Figure 4
To better illustrate the physical and biogeochemical variabilities in this region, different time series should be shown here (not only TChla). For example, T, S, $O_2$, …

Figure 6
This figure does not bring anything. It should be at least merged with an ocean color remote sensing map or something else.

A figure on mooring dataset variability should be included too

---

## Referee Comment (RC2) · Anonymous Referee #2 · 3 Mar 2018

Journal: OS
Title: An integrated open-coastal biogeochemistry, ecosystem and biodiversity observatory of the Eastern Mediterranean. The Cretan Sea component of POSEIDON system.
Author(s): George Petihakis et al.
MS No.: os-2018-3
MS Type: Review article
Special Issue: Coastal marine infrastructure in support of monitoring, science, and policy strategies

General aspects:
The manuscript written by Petihakis et al. describes the components part of the Cretan ecosystem observatory as the coastal and open sea buoys, ferry box, floats, gliders so as the land-based facilities and personnel capabilities. Special interest is given to biochemical measurements and ecosystem modelling. Finally, it gives an overview of future developments in the short term so as in the long one.

The manuscript (hereafter ms) is well written and easy to read. The part concerning the sustainability development is very well addressed, touching important aspects mandatory to maintain the system in the long run, key point of the Observing Systems.

MS deserves to be publishing; however some points I recommend to be reviewed and modified to improve it.

Main parts:

1. Part 2 (A strategic location to study the unknown of the Eastern Mediterranean) should be reduced because it is true that it gives a global idea of the area by touching different aspects but also that it is too discursive and dispersive as regards the real purpose of the ms. I should be synthetized the importance of the Observatory, the role in the East Med, and the benefits the community gains from it measurements. Please reduce the references, it is not a review paper and makes very difficult the read;

2. The Cretan Sea Observatory describes itself as a complex and articulated system. In this context and in MS it is difficult to place it in the Poseidon network. If this is the goal I suggest that the approach used be revised giving a meaning to part 3. The naming of the other two buoys (Pylos and the one in the Athos peninsula) without making any connection between them (Page 9, line 20-25) does not make much sense. Again in this framework, the future vision of the Cretan Sea system would be applied also to the other two buoys?

3. I suggest moving 4.5 after Biochemical Modelling to keep a logical thread in the description of the system components (types of instruments and then

personal and support facilities). Please provide the scheme of the payload of both coastal and open sea buoy.

Specific aspects:

1. Page 8 line 10: please provides some MSFD descriptors as example (useful for those not familiar with the argument)
2. Page 9 line 3: change in ... followed by two other M3A stations in the southern Adriatic (E2) and Ligurian Sea (W1).... This will help to connect this two other sites with what is said in 4.10
3. Page 10 line 18-20: Please rephrase the sentence for better understanding
4. Page 11 (Section 4.7): are any measurements done yet? please indicate an estimate time for the start of the monitoring program
5. Page 13 line 4: the geographical location of the three observatories should be indicated to give more information to the reader.
6. Title 5.2 is not really representative of the section

References:
Please review the reference part since many of them are missing in the text or are in the text and not cited.

Page 3 line 20/28/30 misspelling Theocharis et.al
Correct Cardin et al. 2014 to 2015 in the reference section
Page 4 line 23: Correct Kress et al.2003 to Kress (2003)
Page 5 line 16/line 18: Siokou et al. 2010 ? or Siokou-Frangou et al.2010 or reference is missing?
Page 5 line 39: Turner, 2015? Year does not agree with the reference which state 2002? Misspelling or different paper?
Page 6 line 10: Frangoulis et al. 2005 or 2004? not agree text and reference part
Page 6 line 25: EEA 2015? ---2006???
Page 7 line 26: Ruhl 2001?
Page 9 line 8: Petihakis et al. 2006 --- I'm not sure if Ocean Science Discussion paper can be cited
Page 11 line 26: Baretta et al or Baretta-Bekker?

Missing:
Mentioned in the text and not in the references:
Lascaratos et al. 1999
Theoharis et al. 2009
Pitta and Giannakourou, 2000
Ruhl 2001
Hoteit et al. 2004

In the references but not in the text:
Balopoulos and Collins, 1999
Danovaro et al. 2004
Krom et al. 2004
Mihalopoulos et al. 1997

---

## Referee Comment (RC3) · Anonymous Referee #3 · 8 Mar 2018

The manuscript proposed by G. Petihakis et al. aims giving a detailed overview of the POSEIDON observing system.

The article introduces this observing system for the Eastern Mediterranean sea. Following this aim, no specific scientific question is addressed in the manuscript but it is more dedicated to the description of the POSEIDON objectives and components.

The manuscript is well written and describes briefly each components of the network, including links with ongoing European and international initiatives on Ocean Observing

[Figure]

Systems. A specific focus also appears on the representativeness of the Cretan Sea to monitor Eastern Mediterranean Sea.

As general comments, I suggest that the manuscript would benefit from some illustrations/examples of POSEIDON recent measurements. There are very few examples in the manuscript and it doesn't highlight the potential of such observing system. Furthermore, authors take care of showing the interest of this observatory in this region. It would be interesting to mention how past measurements have contributed to the scientific knowledge of processes in the region.

As the paper aimed to be published in Ocean Science with a wide reading audience, it would be important to be more explicit in acronyms as a lot of projects, initiatives, programs are mentioned in the manuscript. As I'm familiar with this community, I can follow the ideas but for a reader less familiar with those initiatives, it can be difficult to follow in some parts.

Considering the limited needed improvements included in general and specific comments, I recommend this paper for potential publication after minor revisions more related to illustrating more the paper contents.

Please also note the supplement to this comment:
https://www.ocean-sci-discuss.net/os-2018-3/os-2018-3-RC3-supplement.pdf

–––––––––––––––––––––––––––––

**Supplement:**

**General comments**

The manuscript proposed by G. Petihakis et al. aims giving a detailed overview of the POSEIDON observing system.

The article introduces this observing system for the Eastern Mediterranean sea. Following this aim, no specific scientific question is addressed in the manuscript but it is more dedicated to the description of the POSEIDON objectives and components.

The manuscript is well written and describes briefly each components of the network, including links with ongoing European and international initiatives on Ocean Observing Systems. A specific focus also appears on the representativeness of the Cretan Sea to monitor Eastern Mediterranean Sea.

As general comments, I suggest that the manuscript would benefit from some illustrations/examples of POSEIDON recent measurements. There are very few examples in the manuscript and it doesn't highlight the potential of such observing system. Furthermore, authors take care of showing the interest of this observatory in this region. It would be interesting to mention how past measurements have contributed to the scientific knowledge of processes in the region.

As the paper aimed to be published in Ocean Science with a wide reading audience, it would be important to be more explicit in acronyms as a lot of projects, initiatives, programs are mentioned in the manuscript. As I'm familiar with this community, I can follow the ideas but for a reader less familiar with those initiatives, it can be difficult to follow in some parts.

Considering the limited needed improvements included in general and specific comments, I recommend this paper for potential publication after minor revisions more related to illustrating more the paper contents.

**Specific comments**

Abstract
p. 1 / first sentence – The first sentence is mixing as the same kind of "object" processes (air-sea interactions and coastal-open ocean) and parameters. It would be more accurate if authors mentions either processes (physical and bogeochemical) or paramters.

p. 1 / l. 22 – Bio-Argo systems are not measuring Chlorophyll concentrations but fluorescence. I recommend using the latest parameter in the text.

1. International need for observations
p. 2 / l. 11 – EuroGOOS is the first example of the acronyms or notations not defined in the manuscript (as for example and without classification: $C_T$, $A_T$, GCOS, HCMR, NRT, POEM, PELAGOS, …).

2. A strategic location to study the unknowns of the eastern Mediterranean
p. 3 / l. 9 – "… depending on the parameter". I did not understand to which parameter the author is referring to.

3. Aims and mission
p. 7 / l. 4 – The reference to Perivoliotis EuroGOOS extended abstract is misleading as it sounds as it is describing the same content as the present paper. Please consider referring to this abstract for more specific points.

p. 7 / l. 6 – 9 – Reading this part, we wonder about the POSEIDON activity between 2011 and 2017. We understand a bit more in the following sections but it is possible to be more explicit at this stage on this temporal gap ?

p. 7 / l. 29-35 – This "b" topic sounds to me included in "a" topic. Please consider rephrasing those scientific objectives to be more explicit.

4.4 Ferrybox
p. 10 / l. 14-24 – The figure given for FerryBox does not highlight the long term FB activity. Please consider a figure including more track (and recent) for example to illustrate those platforms.

4.7 Gliders
p. 11 / l. 10-15 – A figure showing glider tracks and data would support this subsection in the manuscript.

5.3 Derived biochemical-ecosystem parameters and model state variables
p. 14 / l. 24 – The figure 9 seems very interesting but there is a lack of explanation in the manuscript. Please consider adding more information on this figure or if you consider that it would need too much details, you can consider removing this figure and replacing by model results examples.

6. Metadata and data handling
p. 15 / l. 1 – Are ADCP and sediment traps available on a different portal ?

**Minor and technical corrections**
1. International need for observations
p. 2 / l.5 – "ocean bottom" could be replace by "bottom ocean"

p. 2 / l.15 and l.22 – In the manuscript, both terms are used: "biochemical" and "biogeochemical". Even if those terms are used often without distinction, this is not the same definition. Please, consider to be more homogeneous in the whole manuscript.

p. 2 / l. 24 – Authors cite "recently" for a study from 2015. Please consider rephrasing the introduction of the sentence.

2.1 Coupling of biochemistry with circulation patterns
p. 3 / l. 34 – To keep the paper understood for the next century, I would suggest to use "1970s" instead of "70s".

2.4 Basin to global anthropogenic impact
p. 6 / l. 25 – The reference to EEA is 2006 and not 2015 in the reference list. Please double check the reference.

4. Components-Platforms
p. 8 / l. 20-31 – A reference to the map in Figure 1 is missing in this section introduction.

Tables
Table 1
+ pCO2 and ADCP are missing in the list.
+ Please consider adding the sampling frequency range for each parameter.
+ The sensor replacement frequency would also benefit readers interested in managing an observing system.

Figures

Figure 1

+ The figure 1 is a key figure for the paper. I think that a green less dark to highlight the are would help to see other system locations.

+ Please add a depth scale/colorbar

Figure 3

+ this figure is tricky to read. Please consider splitting in 4 subplots (cruises, FB, buoy, sed trap + ADCP) for clarity.

---

## Author Comment (AC4) · 25 May 2018

The additional or modified figures requested by the reviewers

[Figure]

**Fig. 1.**

[Figure]

**Fig. 2.**

[Figure]

**Fig. 3.**

[Figure]

**Fig. 4.**

[Figure]

**Fig. 5.**

[Figure]

**Fig. 6.**

---

## Author Comment (AC6) · 25 May 2018

The additional or modified figures requested by the reviewers

[Figure]

[Figure]

**Fig. 1.** Figure 1

[Figure]

**Fig. 2.** Figure 3

[Figure]

**Fig. 3.** Figure 4

[Figure]

**Fig. 4.** Figure 5

[Figure]

**Fig. 5.** Figure 6

[Figure]

**Fig. 6.** Figure 8

[Figure]

**Fig. 7.** Figure 9

---

## Author Response (AR1)

**Replies to Reviewers**

We would like to thank all three reviewers for their constructive comments that helped improve our manuscript.
Further we give our response to the comments:

**Replies to Reviewer 1**

General comments
The present manuscript aims to describe the current POSEIDON observing system (OS) in the East Mediterranean Sea which is based on multi-platforms and multi-parameters approach. The actual status of this OS is composed by coastal and open sea eulerian and lagrangian platforms (gliders, Argo floats, Ferrybox, moorings and buoys). This OS delivers variables which are essential to understand and predict the impacts of climate change and anthropogenic pressure on marine ecosystems in the Eastern Mediterranean Sea.
The manuscript is well written but some parts are missing and some parts should be re-organized.

The section 2 is too long and it should be reduced. The authors should do a rapid state of arts of the situation in the eastern basin and describe the main characteristics of the basin (why is it important to implement and sustain an integrated OS in this region).

Reply 1:
Section 2 was reduced by approximately 40%, with emphasis on the Eastern Mediterranean in relation to the importance of implementing and sustaining an integrated OS in this area.

In the section 3, the scientific questions, which are the backbone of each OS, should be detailed in the first part followed by the management aims and services.

Reply 2:
The requested change was made.

In the section 4, the personnel section should be moved after the section 4.9 otherwise this is confusing to see the personnel skills between the description of Ferrybox and Argo floats.

Reply 3:
The requested change was made.

In this section, there is no description of the POSEIDON buoys used for forecasting products. Why ?

Reply 4:
As a reply to this question in section 4.7 the following text was added (p.10 lines 32-37).

"Data from POSEIDON buoys such as E1-M3A have been extensively used for model validation/calibration and testing of model parameterization techniques adopted in the operational POSEIDON models. Although the assimilation of these data directly in the model forecasts would have a relatively limited effect, given their small spatial coverage, they are of paramount importance for the development and testing of data assimilation schemes, as well as in the analysis of specific processes and the underlying dynamics of the system."

After the data description in the section 5, a section on data analysis and data quality control is largely missing: How real-time data are managed ? Which protocols are used ? How the delayed mode data are adjusted ? Is there different correction levels (as for the remote sensing data) ? For example in Tables 1, 2 & 3, the authors should add some information on correction methods applied for each variables. In the same way, accuracy and measurements frequency should be added in these tables.

Reply 5:
The section 6 "Metadata and data handling" was rewritten to include the information requested. Accuracy was added in all tables. Frequency of measurements was added as a separate column in Table 1. In tables 2 and 3 frequency was described in the table title.

An additional section on the POSEIDON management and governance is missing too: who decide the choice of variables ? sensors ? how the new scientific needs are taken into account ? for example nowadays, global OS are focusing more and more on biological variables for marine ecosystems change (e.g. genomics). What is the vision of the POSEIDON group on this question ?

Reply 6:
The section 7, now entitled "Management, governance and sustainable development" was rewritten to include the information requested.

Regarding the scientific production, which is depending on the POSEIDON OS sustainability, how many publications, thesis, reports have been produced ? Any information regarding the DOI dataset statistics should be also included (at least on annex) to demonstrate the importance to maintain such OS.

Reply 7:
At the last paragraph of section 7 in p.16 line 4, the number of peer review publications, conference presentations and PhD thesis was added.

Finally, a description and figure on integrated results from the different POSEIDON components are missing. This should prove the necessity to implement an integrated and multi-variables OS but also it will stress what are the actual gaps and needs.

Reply 8:

Integrated results are now presented in the new figures 8 and 9, which are discussed in sections 4.4 and 4.6 respectively.

Ancillary comments
The word "parameter" is often over-interpreted. A distinction with variable should be addressed here. Parameters are not natural quantities, variables are. A variable is an entity that changes over time or depth: this what we are measuring. In general, parameters are "constants" that define a specific instance of a general equation that is based on variables. That's why we used the term EOV for Essential Oceanic Variables.

Reply 9:
We agree with the reviewer on the use of those two words. The term "parameter" was changed to "variable" when appropriate.

"Biochemical" should be changed to "biogeochemical"

Reply 10:
The change was made in the entire manuscript

Figure 1 Information is missing regarding the glider endurance lines, the Argo floats observation area, the sediment traps deployment, etc... A new figure should be proposed with the names of seas, straits, main countries, ...

Reply 11:
The glider endurance line was added in addition to the geographical location of other observatories requested by reviewer 2 (see reply 24). The sediment traps location is the same as the E1-M3A buoy (as described in section 4.2 and in the new modified legend of Figure 1). The Argo floats observation area was not added as it is highly variable. Names of seas cited in the manuscript were added in the figure. We believe adding names of countries too will complicate the figure and remove the focus from the observing systems.

Figure 3 This is not very clear and too small. It should be modified or removed

Reply 12:
The figure was modified by being split in subplots as suggested by reviewer 3 (see reply 51)

Figure 4 To better illustrate the physical and biogeochemical variabilities in this region, different time series should be shown here (not only TChla). For example, T, S, $O_2$, ...

Reply 13:
Temperature and salinity were also added to in the Figure (see new figure 6).

Figure 6 This figure does not bring anything. It should be at least merged with an ocean color remote sensing map or something else.

Reply 14:
Ocean color data were added as well as more and recent FB tracks (see new figure 8) as requested by reviewer 3. A description of the new figure was added in section 4.4

A figure on mooring dataset variability should be included too

Reply 15:
Mooring dataset variability is shown in the new figure 4

**Replies to Reviewer 2**

General aspects:
The manuscript written by Petihakis et al. describes the components part of the Cretan ecosystem observatory as the coastal and open sea buoys, ferry box, floats, gliders so as the land-based facilities and personnel capabilities. Special interest is given to biochemical measurements and ecosystem modelling. Finally, it gives an overview of future developments in the short term so as in the long one.
The manuscript (hereafter ms) is well written and easy to read. The part concerning the sustainability development is very well addressed, touching important aspects mandatory to maintain the system in the long run, key point of the Observing Systems.
MS deserves to be publishing; however some points I recommend to be reviewed and modified to improve it.
Main parts:
   1.  Part 2 (A strategic location to study the unknown of the Eastern Mediterranean) should be reduced because it is true that it gives a global idea of the area by touching different aspects but also that it is too discursive and dispersive as regards the real purpose of the ms. I should be synthetized the importance of the Observatory, the role in the East Med, and the benefits the community gains from it measurements. Please reduce the references, it is not a review paper and makes very difficult the read;

Reply 16:
This part was reduced. See Reply 1.
Approximately 25 references were also removed.

   2.  The Cretan Sea Observatory describes itself as a complex and articulated system. In this context and in MS it is difficult to place it in the Poseidon network. If this is the goal I suggest that the approach used be revised giving a meaning to part 3. The naming of the other two buoys (Pylos and the one in the Athos peninsula) without making any connection between them (Page 9, line 20-25) does not make much sense. Again in this framework, the future vision of the Cretan Sea system would be applied also to the other two buoys?

Reply 17: In the first paragraph section 3 the Cretan Sea Observatory was better defined, as a subsystem of the Poseidon network.

The last paragraph of section 4.1 p. 8 lines 13-24 was rewritten to show the connection between buoys.
A sentence was added in section 8 p. 16 lines 21-22 to show the vision concerning the Pylos and Athos Buoys.

3. I suggest moving 4.5 after Biochemical Modelling to keep a logical thread in the description of the system components (types of instruments and then personal and support facilities).

Reply 18:
The requested change was made.

Please provide the scheme of the payload of both coastal and open sea buoy.

Reply 19:
A new figure (Figure 3) was added showing the scheme of the payload of both coastal and open sea buoy.

Specific aspects:
1. Page 8 line 10: please provides some MSFD descriptors as example (useful for those not familiar with the argument)

Reply 20:
Some MSFD descriptors were added as an example (p.7 lines 2-3).

2. Page 9 line 3: change in ... followed by two other M3A stations in the southern Adriatic (E2) and Ligurian Sea (W1).... This will help to connect this two other sites with what is said in 4.10

Reply 21:
The requested change was made (p.7 lines 33-34).

3. Page 10 line 18-20: Please rephrase the sentence for better understanding

Reply 22:
The sentence was rephrased (p.9 lines 15-17).

4. Page 11 (Section 4.7): are any measurements done yet? please indicate an estimate time for the start of the monitoring program

Reply 23:
Time of the start of the Glider program was added in the text (p.10 line 5), and a new related figure (Fig. 9) was added. The Glider endurance line was added in Figure 1.

5. Page 13 line 4: the geographical location of the three observatories should

be indicated to give more information to the reader.

Reply 24:
The geographical location of the three observatories was added in Figure 1

6. Title 5.2 is not really representative of the section

Reply 25:
The title was modified to be more representative of the section

Reply 26:
References were checked and corrected

**Replies to Reviewer 3**

The manuscript proposed by G. Petihakis et al. aims giving a detailed overview of the POSEIDON observing system. The article introduces this observing system for the Eastern Mediterranean sea. Following this aim, no specific scientific question is addressed in the manuscript but it is more dedicated to the description of the POSEIDON objectives and

components. The manuscript is well written and describes briefly each components of the network, including links with ongoing European and international initiatives on Ocean Observing Systems. A specific focus also appears on the representativeness of the Cretan Sea to monitor Eastern Mediterranean Sea.

As general comments, I suggest that the manuscript would benefit from some illustrations/examples of POSEIDON recent measurements. There are very few examples in the manuscript and it doesn't highlight the potential of such observing system.

Reply 27:
New figures were added including recent measurements: Figure 4, Figure 6, Figure 8, Figure 9

Furthermore, authors take care of showing the interest of this observatory in this region. It would be interesting to mention how past measurements have contributed to the scientific knowledge of processes in the region.

Reply 28:
An example of contribution of past measurements is shown in the new Figure 4

As the paper aimed to be published in Ocean Science with a wide reading audience, it would be important to be more explicit in acronyms as a lot of projects, initiatives, programs are mentioned in the manuscript. As I'm familiar with this community, I can follow the ideas but for a reader less familiar with those initiatives, it can be difficult to follow in some parts.

Reply 29:
All acronyms were checked as to be cited full the first time

Considering the limited needed improvements included in general and specific comments, I recommend this paper for potential publication after minor revisions more related to illustrating more the paper contents.

Specific comments
Abstract
p. 1 / first sentence – The first sentence is mixing as the same kind of "object" processes (air-sea interactions and coastal-open ocean) and parameters. It would be more accurate if authors mentions either processes (physical and bogeochemical) or paramters.

Reply 30:
The sentence was rephrased (p.1 lines 12-14)

p. 1 / l. 22 – Bio-Argo systems are not measuring Chlorophyll concentrations but fluorescence. I recommend using the latest parameter in the text.

Reply 31:

The sentence was adapted to specify that, for Bio-Argo and other sensors, fluorescence is measured as a proxy of Chlorophyll-a (p.1 lines 23-24).

1. International need for observations
p. 2 / l. 11 – EuroGOOS is the first example of the acronyms or notations not defined in the manuscript (as for example and without classification: CT, AT, GCOS, HCMR, NRT, POEM, PELAGOS, ...).

Reply 32:
See Reply 29

2. A strategic location to study the unknowns of the eastern Mediterranean
p. 3 / l. 9 – "... depending on the parameter". I did not understand to which parameter the author is referring to.

Reply 33:
The sentence was rephrased (p.3 lines 8-9).

3. Aims and mission
p. 7 / l. 4 – The reference to Perivoliotis EuroGOOS extended abstract is misleading as it sounds as it is describing the same content as the present paper. Please consider referring to this abstract for more specific points.

Reply 34:
A sentence was added at the beginning of section 3 (p. 5 lines 26-27) to distinguish the CS observatory as a subsystem of POSEIDON system thus specifying that the reference Perivoliotis et al. includes the CSO subsystem.

p. 7 / l. 6 – 9 – Reading this part, we wonder about the POSEIDON activity between 2011 and 2017. We understand a bit more in the following sections but it is possible to be more explicit at this stage on this temporal gap ?

Reply 35:
A sentence was added (p. 5 lines 33-35) to specify activities between 2011 and 2017.

p. 7 / l. 29-35 – This "b" topic sounds to me included in "a" topic. Please consider rephrasing those scientific objectives to be more explicit.

Reply 36:
The topics "a" and "b" were rephrased.

4.4 Ferrybox
p. 10 / l. 14-24 – The figure given for FerryBox does not highlight the long term FB activity. Please consider a figure including more track (and recent) for example to illustrate those platforms.

Reply 37:
See Reply 14

4.7 Gliders
p. 11 / l. 10-15 – A figure showing glider tracks and data would support this subsection in the manuscript.

Reply 38:
See Reply 23

5.3 Derived biochemical-ecosystem parameters and model state variables
p. 14 / l. 24 – The figure 9 seems very interesting but there is a lack of explanation in the manuscript. Please consider adding more information on this figure or if you consider that it would need too much details, you can consider removing this figure and replacing by model results examples.

Reply 39:
More information concerning the figure 12 (ex figure 9) was added in section 5.3 p.14 lines 8-14

6. Metadata and data handling
p. 15 / l. 1 – Are ADCP and sediment traps available on a different portal ?

Reply 40:
Unfortunately, ADCP and sediment trap data are not available yet, something that we are working and hope it will be solved in the near future.

Minor and technical corrections
1. International need for observations
p. 2 / l.5 – "ocean bottom" could be replace by "bottom ocean"

Reply 41:
The replacement was made.

p. 2 / l.15 and l.22 – In the manuscript, both terms are used: "biochemical" and "biogeochemical". Even if those terms are used often without distinction, this is not the same definition. Please, consider to be more homogeneous in the whole manuscript.

Reply 42:
See Reply 10

p. 2 / l. 24 – Authors cite "recently" for a study from 2015. Please consider rephrasing the introduction of the sentence.

Reply 43:

The sentence was rephrased. In addition, the term "recently" was removed in most locations in the text are the year was kept only.

2.1 Coupling of biochemistry with circulation patterns
p. 3 / l. 34 – To keep the paper understood for the next century, I would suggest to use "1970s" instead of "70s".

Reply 44:
The term "70s" no longer exists as the corresponding phrase was removed due to the requested reduction of section 2.

2.4 Basin to global anthropogenic impact
p. 6 / l. 25 – The reference to EEA is 2006 and not 2015 in the reference list. Please double check the reference.

Reply 45:
The reference was checked and corrected

4. Components-Platforms
p. 8 / l. 20-31 – A reference to the map in Figure 1 is missing in this section introduction.

Reply 46:
A reference to Figure 1 was made in this section

Tables
Table 1
+ pCO2 and ADCP are missing in the list.

Reply 47:
pCO2 and ADCP were added in the Table 1

+ Please consider adding the sampling frequency range for each parameter.

Reply 48:
The sampling frequency of each parameter was added in Table 1

+ The sensor replacement frequency would also benefit readers interested in managing an observing system.

Reply 49:
The sensor replacement frequency was added in section 5.1

Figures
Figure 1

+ The figure 1 is a key figure for the paper. I think that a green less dark to highlight the are would help to see other system locations.
+ Please add a depth scale/colorbar

Reply 50:
The requested changes were made

Figure 3
+ this figure is tricky to read. Please consider splitting in 4 subplots (cruises, FB, buoy, sed trap + ADCP) for clarity.

Reply 51:
The figure was split in 3 subplots (cruises, buoy+FB, sed trap+ADCP)

[revised manuscript text omitted]

**3 Aims and mission**

The Cretan Sea Observatory, is the most biogeochemistry/ecosystem oriented, multiplatform subsystem of POSEIDON. POSEIDON (www.poseidon.hcmr.gr) is an observatory research infrastructure of the Eastern Mediterranean basin, for the monitoring and forecasting of the marine environment, supporting the efforts of the international and local community and replying to the needs and gaps of science, technology and society (Perivoliotis et al., 2018). It was developed in three phases under the funding of EEA Financial Mechanism (85%) and Greek National funds (15%): POSEIDON-I (1997-2000), a first-generation buoy monitoring network with operational centre, forecasting system, and relevant human resources; POSEIDON-II (2005-2008) a system upgrade and expansion; and finally POSEIDON-III (2009-2011) a deep sea observing capacity expansion.

During the period 2011 to 2017 they were no major upgrades and the system was partly supported by EU and National Projects. In 2017, an extended upgrade and renewal of POSEIDON buoy network monitoring parts and components and their supporting hardware was realized with the implementation of an integrated marine monitoring program funded by the EEA Financial Mechanism 2009-2014. The POSEIDON's biogeochemistry-ecosystem observational component aims, mainly achieved via its most oriented biogeochemistry-ecosystem oriented subsystem of the Cretan Sea, are presented below.

.a. Maintain a quality controlled biochemical ecosystem database, with reduced data delivery time and expanded accessibility (see Sect. 6)

.a. Support ecosystem model validation for improved ecosystem forecasting and management

.a. Provide a technology test bed for new biogeochemical sensors

.a. Maintain collaboration through strong networking with similar observatories for common strategy, complementary tasks, protocol standardisation, and transfer of knowledge (see Sect. 4)

H.A. A.    The **scientific objectives** coming out from the international (Ruhl, 20112001; IMBER, 2005, 2010; NRC, 2011), European (EGMRI, 2013) as well as national scientific projects, are to provide scientific knowledge and support on the study of:

a. ocean mechanisms, including their interactions and spatiotemporal variability (duration, occurrence, match-mismatches). The currently examined mechanisms are: solubility and biological pump, transformation and transfer of matter (fluxes within food webs and in-out of ocean interfaces), food web structure (stock, functional groups, microbial loop, size spectra) and ocean-atmosphere interactions (heat, dust, $CO_2$).

b. food web structure (stock, functional groups, microbial loop, size spectra) and biodiversity related variablesunknowns (number of species present and their temporal variability), in order to help understand species interaction with food web functioning and biogeochemical cycling.

c. the sensitivity of biodiversity and the variability of ocean mechanisms in relation to combined natural forcing factors and anthropogenic pressures. The current pressures considered, are those induced by warming (temperature, circulation, stratification, wind mixing, extreme-episodic events), ocean acidification (carbonate chemistry), nutrient dynamics (stoichiometry, dust deposition) and oxygen concentration. Other pressures and feedback effects planned to be considered in the future studies of the observatory are presented in the Sect. 8.

B.   B.      The **Management aims** based on the experience acquired by POSEIDON team and international management recommendations (NRC, 2011; Karl, 2010; Ruhl et al., 2011) are:

a.   Sustainability through prioritization and cost-effective management. This aim guides the expansion of the number and of the spatiotemporal coverage of the variables studied (see Sect. 7)

a.   Integration of disciplines, platforms and analysis methods, by bridging various measurement scales and methodologies as well as through a step-by-step integration of an increasing number of different biogeochemical platforms (see Sect. 4), variables and analysis methods (see Sect. 5)

b.   Maintain a quality controlled biogeochemical-ecosystem database, with reduced data delivery time and expanded accessibility (see Sect. 6)

c.   Support ecosystem model validation for improved ecosystem forecasting and management

d.   Provide a technology test bed for new biogeochemical sensors

e.   Maintain collaboration through strong networking with similar observatories for common strategy, complementary tasks, protocol standardisation, and transfer of knowledge (see Sect. 4)

C.   Provide services to marine **policy-makers and the society,** while adaptation to the evolving needs is ensured by a feedback mechanism put in place. POSEIDON is being developed in accordance to the policy frameworks suggested by IOC/GOOS, EuroGOOS, the Mediterranean Operational Network for the Global Ocean Observing System (MonGOOS) and the Group on Earth Observations (GEO). The achievement of

these objectives is made through collaboration with scientists from other disciplines including social scientists, and in communication with policy makers and the wider public. Within this framework the observatory aims to:

   a.f. Provide policy makers with sound scientific knowledge to assist in making informed decisions. The observatory provides several MSFD descriptors (D1: Biological Diversity, D2: Commercial Fish, D3: Food web, D4: Hydrological conditions) and tests indicators of ecosystem health status.

   b.g. Promote understanding of anthropogenic impact on the ocean systems, and at the same time the dependence on them.

   c.h. Provide products to the end user through the POSEIDON operational biogeochemistrybiochemistry and ecosystem modelling tools.

[revised manuscript text omitted]

Since 2017 two more POSEIDON buoys outside the Cretan Sea, also provide biogeochemical data ($O_2$, Chl-a). These two similar to E1-M3A, Oceanor Wavescan buoys are deployed in the N. Aegean and Ionian waters.  The buoy placed between Athos peninsula and the island of Lemnos (AB) (Fig. 1), monitors an area affected (e.g. with increased productivity) by the Black Sea water entering the North Aegean through the Dardanelles straits, which plays a significant role modulating the thermohaline characteristics and dynamics of the whole Aegean Sea, including the Cretan Sea (e.g. Korres et al., 2014). The Pylos site (PB) in the SE Ionian is a crossroad where intermediate and deep-water masses meet. The site is located on the pathway of the Aegean Sea dense water that travels to the north along the western coast of Greece. These three buoys (E1-M3A, AB and PB) providing meteorological, physical, and biogeochemical data in different areas, have allowed comparison of trends of physical variables (such as temporal increasing trend in temperature, see Fig. 4) and may allow the future comparison of biogeochemical variables in different levels of oligotrophy.

**4.2 Water column sampling and sediment traps (R/V cruises). From occasional high spatial resolution snapshots to sustained temporal coverage**

[revised manuscript text omitted]

The gliders' T/S profiles, collected during a 6-month period, will be used in Observing Systems Experiments (OSE) to assess the impact of the continuous monitoring of the Cretan Sea to the hydrodynamic modelling of the Aegean Sea, as the former plays a significant role in water masses exchanges with the Eastern Levantine basin, through the east and west Cretan Straits. It is interesting to see that the sea-surface height distribution as derived from satellite altimetry (Fig. 9a) is reasonably consistent with the vertical temperature and salinity structures depicted in Figure 9b -9c. In fact, as the glider transverses from west to east, the elongated anticyclonic eddy structure located to the north of Crete at approximately 23.8ºE - 24.8ºE (Fig. 9a) displays a noticeable deepening of the isothermals and isohalines, as revealed by the measured profiles that goes down to 300 m depth. It is expected that the introduction of these glider measured profiles to the Aegean Sea dynamics,

through data assimilation system, will act synergistically with the satellite altimetry trivially assimilated into the system.

**4.9 4.7 Biogeochemical Modelling**

Forecasting tools are centrally placed in the POSEIDON system, with a number of state-of-the-art weather, wind waves, ocean circulation and marine ecosystem numerical models, initialization and data assimilation schemes providing 5-days ahead information on daily basis regarding the atmospheric (Papadopoulos et. al., 2002), sea state (Korres et., 2011) and hydrodynamic conditions (Korres et al., 2010) in the Aegean/Ionian Seas and in the Mediterranean. Currently, the POSEIDON modelling group is providing the wave forecasting products of the Copernicus Marine Environment Monitoring Service (CMEMS) for the Mediterranean Sea in the framework of MED-MFC.

The POSEIDON ecosystem simulation tool is one of the first developed in the Mediterranean, producing daily forecasts for a range of ecosystem variablesparameters for the whole basin. The Cretan Sea has been a test site for the implementation of the biogeochemical European Regional Seas Ecosystem Model (ERSEM, Baretta-Bekker et al., 1995), since it is the only offshore site in Greece in which the ecosystem has been systematically observed, providing a very successful test bed for model development (Petihakis et al., 2002; Triantafyllou et al., 2003a, b, c; Hoteit et al., 2004). Data from POSEIDON buoys such as E1-M3A have been extensively used for model validation/calibration and testing of model parameterization techniques adopted in the operational POSEIDON models. Although the assimilation of these data directly in the model forecasts would have a relatively limited effect, given their small spatial coverage, they are of paramount importance for the development and testing of data assimilation schemes, as well as in the analysis of specific processes and the underlying dynamics of the system.; 
[revised manuscript text omitted]
 (e.g. dinoflagellates >20 μm and <20 μm) and/or trophic functioning (e.g. autotrophic and heterotrophic nanoflagellates). Mixotrophs data field data were the most particular case, since they were split in four subgroups, creating an autotrophic and a heterotrophic subgroup to express mixotrophy (not represented in the model), which were subdivided again based on cell size (>20 μm and <20 μm). A model application using such field data grouping/splitting for validation can be found in Tsiaras et al. ( 2017). Planktonologists and the modellers of the

observatory agreed that there is no rule of thumb, and model-field data correspondences should be adjusted according to the season and region.

**6    Metadata and data handling**

 The POSEIDON database is set to include BGC and BGC-associated variables, either remotely sensed (pH, pCO$_2$, Chl-a, O$_2$, meteorological, T, S), or obtained from in-situ sampling (Chl-a, O$_2$, nutrients, plankton stock). The processing and the Quality Control procedures for the data collected from all the  POSEIDON stations comply with the Copernicus Marine Environment Monitoring Service (CMEMS) Insitu Thematic Assembly Center (TAC) procedures, as HCMR is the regional data distribution node for the Mediterranean Sea. A number of quality control procedures for the validity of the data and a series of metadata correctness tests are applied before the release of the relevant data files. The data quality control process includes different routines for Near Real-Time (NRT) products  and the Delayed Mode/Reprocessed Products.

The Near real Time quality control consists of a set of automatic tests according to the EuroGOOS Data Management, Exchange and Quality Working Group (DATA-MEQ) recommendations (http://eurogoos.eu/data-management-exchange-quality-working-group-data-meq/). These procedures are defined by variable, elaborated in coherence with international agreements, in particular those adopted within SeaDataNet  project (https://www.seadatanet.org/Standards/Data-Quality-Control) and they are applied by all the regional nodes of CMEMS Insitu TAC on the NRT products in order to assure a minimum level of quality. Detailed information on the applied procedures can be found in the CMEMS INSTAC QC procedure manuals for temperature and salinity, currents and sea level (http://dx.doi.org/10.13155/36230), waves (http://doi.org/10.13155/46607), Chl-a (fluorescence), dissolved oxygen and nutrients (http://doi.org/10.13155/36232 ).

During the Delayed Mode/Reprocessed data analysis, procedures assessing the consistency of the data over a period of time are applied to the time series. The scientific validation includes statistical tests to check the consistency of the observations and climatological tests to highlight suspicious data that could not be detected by the automatic Quality Control processes. The resulted outliers are reassessed through visual inspection, a procedure that has an increased level of complexity on its implementation and highly relies on scientific expertise. Delayed Mode/Reprocessed products are available for temperature, salinity and waves, while the relevant product for Chl-a (fluorescence) and dissolved oxygen is under development.

 Data can be visualized through the POSEIDON web-site (fixed platforms, Ferrybox) and the MonGOOS data portal (http://www.mongoos.eu/data-center, all platforms except sediment traps, ADCP), while the data are freely available to the public, the stakeholders and the scientific community, acknowledging the EC INSPIRE directive, in order to enable easy access to data and their reuse.

**7   Management, governance and sustainableSustainable development**

POSEIDON, and thus its subsystem of the Cretan Sea Observatory, has been managed up to now at institutional level by the HCMR Operational Oceanography Unit (i.e. POSEIDON team) which is subdivided in four components, three headed by scientists (observing, forecasting/modelling, data center) and one headed by an engineer (technical component). Within this management, the choice of infrastructures, sensors and thus BGC variables has been mainly driven by the participation in EU projects (e.g. JERICO, FixO3, JERICO-NEXT) which follow EU and International priorities. The recent emphasis given to biogeochemistry has been mainly the outcome of participation to JERICO and JERICO-NEXT. The latest project will allow to the Cretan Sea Observatory to integrate in the nearby future new biogeochemical variables by following the recent developments in sensors observing carbonate system variables, pollutants, phytoplankton and microbial diversity, toxic algae, etc.

In the near future the management/governance will move to a national level, as in 2018 the Hellenic Integrated Marine and Inland Observing Forecasting and Offshore Technology Systems Observing and Forecasting System (HIMIOFOTS), a national research marine observing infrastructure has been initiated. Within the HIMIOFOTS network a national management/governance is planned subdivided in (i) coordination team, (ii) operational and development team, (iii) scientific/technological committee and (iv) advisory team for infrastructure users. These boards will supervise the execution of the infrastructure's strategic plan, the scientific excellence, the technological development/innovation, the long-term sustainability, the good access to the infrastructure by national and international scientists, the outreach activities and the participation in large research international infrastructure networks.

A critical issue for any observatory is its sustainability. A continuous funding which will allow not only the day-to-day operations, but also the upgrade to the current state of the art technology is crucial. Unfortunately, in most cases marine observatories in Europe are developed through intermittent funding and national incentives. Likewise, the fixed platform of the Cretan Sea's observatory was founded through the FP6 EU Mediterranean Forecasting System Pilot Project (MFSPP) followed by the POSEIDON project and the EFTA funds. Furthermore, some activities and developments have been funded in the framework of both EU and national projects (research and infrastructural), while in 2018recently the observatory became part of the  (HIMIOFOTS.), a research infrastructure  part of the Greek Research Infrastructure road map.

In periods when funds are limited, it is important to have and maintain a baseline via prioritisation of the variablesparameters observed and platforms used. Such a plan must take into consideration, among other, the existing historical data sets, the international priorities and efforts and the specific scientific questions in the wider area (Eastern Mediterranean).

Despite some periods of low support, the multivariablemultiparameter-multiplatform  approach and the resulting scientific production (>80 peer reviewed publications, >170 conference presentations, 7 PhD thesis) allowed, allows the participation to various targeted research projects and thus the provision of funds through multiple sources. In addition, the long experience in the entire chain of operations and the particular conditions of the Eastern Mediterranean make the observatory an excellent test bed both for new technology and sampling methods.

**8 Future scopes, expansion and vision**

The future of the observatory is presented in a short-term strategy and a long-term vision.

1. The short-term strategy of the biogeochemical observatory follows the expansion vision of POSEIDON system which considers recommendations, guidelines and priorities defined in the national Research
5 Infrastructure road map of observing systems (HIMIOFOTS), review papers (e.g. Claustre et al., 2010), EU goals directives (MSFD, H2020), and visions of European (EMB and EuroGOOS, e.g. EGMRI, 2013) and International coordinating bodies (Global Ocean Observing system - GOOS, and Global Climate Observing System - GCOS).

A main short-term goal is attaining a NRT character for the biogeochemical variables together
10 with a further expansion of the recorded variables, with a greater focus in air-sea interaction. Based on the key variables recommended by the EU (EGMRI, 2013), priority is currently given to further integrate sensors of $O_2$, $CO_2$, pH and fluorescence (Chl-a). Nutrient sensing is expected to be the next to follow, although the low concentrations found in the Eastern Mediterranean constitute a strong technological challenge. HCMR plans to expand the ability to host such biogeochemical sensors (with NRT capability) to more
15 of the existing POSEIDON platforms (e.g. buoys, Bio-Argos, gliders, drifters, Ferrybox) beyond the Cretan Sea, as done in 2017 with the addition of biogeochemical sensors to the NE Aegean and SE Ionian Sea buoy. 
[revised manuscript text omitted]

| | Ferrybox | Thermo-Salinometer SBE 45 | ± 0.005 °C / ± 0.0005 S/m | 1min |
| | RV (CTD) | SBE 19+ OR SBE911 | ± 0.005 °C / ± 0.0005 S/m | 1 sec |
| | Glider | SBE GPCTD (Glider Payload CTD) | ± 0.002 °C / ± 0.0003 S/m | 30 sec |
| Fluorescence (Chl-a) | Buoys (E1-M3A, AB, PB) | Wetlabs FLNTU | 0.025 µg/L Chl | 180 min |
| | Ferrybox | Scufa II Turner Design | 0.02 µg/L Chl | 1 min |
| | RV (CTD) | WET Labs ECO-AFL/FL 9 or Chelsea Aqua 3 | 0.025 µg/L Chl | 1 sec |
| Dissolved Oxygen | Buoys (E1-M3A, AB, PB) | SBE43 / SBE 63/ Aanderaa Optode | ± 2% / ± 2%/ ± 5% | 180 min |
| | Ferrybox | Aanderaa optode | ± 5% | 1 min |
| | Bio-Argos | Aanderaa optode | ± 5% | 1 min |
| | RV (CTD) | SBE 43 | ± 2% | 1 sec |
| | Glider | SBE 43F | ± 2% | 30 sec |
| Turbidity | RV (CTD) | WET Labs ECO FLNTU | 0.013 NTU | 1 sec |
| PAR/Irradiance | RV (CTD) | Biospherical/Licor | N/A | 1 sec |
| pH | Buoy (E1-M3A) | Sensor LabpH | ± 0.005 pH units | 180 min |
| | Ferrybox | Meinsberg probe | ± 0.3 pH units | 1 min |
| pCO$_2$ | Ferrybox | ControsCO2 | ± 0.5% | 1 min |
| | Buoy (E1-M3A) | Pro-Oceanus | 2 ppm | 180 min |
| ADCP (backscattering) | Buoy (E1-M3A) | Teledyne RDI 75kHz | ± 1% | 180 min |

**Table 2: Variables measured from discrete bottle and net samples at high frequency (monthly). Method ranking from Lorenzoni and Benway (2013).**

| Variable | Platform | Analytical method | Method Ranking | Accuracy |
|---|---|---|---|---|
| $NO_3+NO_2$, $Si(OH)_4$ | E1-M3A, HCB | Manual Spectrophotometric | Acceptable | <3%
 <4% |
| $PO_4$ | E1-M3A, HCB | Magnesium-induced co-precipitation | Best | <2% |
| Total Chl-a | E1-M3A, HCB | Fluorescence and HPLC | Best | NA |
| Other Phytopigments | E1-M3A, HCB | HPLC | Best | NA |
| Viruses and Bacteria | E1-M3A | Flow cytometry | Best | NA |
| Picophytoplankton | E1-M3A | Flow cytometry | Best | NA |
| Nanophytoplankton | E1-M3A, HCB | Microscopy (UV + blue light excitation) | Best | NA |
| Other nanoplankton | E1-M3A, HCB | Inverted microscopy | NA | NA |
| Microphytoplankton | E1-M3A, HCB | Inverted microscopy | Best | NA |
| Ciliates | E1-M3A | Inverted epifluorescence microscope (blue light excitation) | NA | NA |
| Zooplankton | E1-M3A, HCB | 45μm and 200 μm nets, Scanning & Image analysis | Best | NA |

**Table 3: Variables measured from discrete bottle samples at low frequency (6 to 24 months) and from sediment traps (integrating 15 days). Method ranking from Lorenzoni and Benway (2013). +traps : measurement from both water column samples and from particulate matter in sediment traps. FB: Ferrybox.**

| Variable | Platform | Analytical method | Method Ranking | Accuracy |
|---|---|---|---|---|
| Dis. Oxygen | E1-M3A, HCB | Winkler (against CTD sensor) | Best | $\leq$0.02 mL/L |
| $A_T$ | E1-M3A, FB | Potentiometric titration (Closed cell) | Best | $\leq$0.1% |
| DIC | E1-M3A, FB | Coulometric determination | Best | $\leq$0.1% |
| TEP | E1-M3A | Colorimetric determination | NA | $\leq$11% |
| POC/ PN | E1-M3A (+ traps) | High Temperature Combustion via Elemental Analyzer | Good | $\leq$3.5% / $\leq$2.5% |
| DOC or TOC | E1-M3A | High Temperature Combustion | Best | $\leq$3% |
| TDN | E1-M3A | Persulfate Oxidation | Good | $\leq$5% |
| DOP | E1-M3A | Persulfate Oxidation | Best | $\leq$5% |
| Primary production | E1-M3A | 14C. Fractional day incubations scaled to daily rates | Acceptable | NA |
| Bacterial production | E1-M3A | [3]H-labelled leucine method | Best | NA |

[Figure]

[Figure]

**Figure 1: Map of showing the location of the Cretan Sea (green ellipse) and all POSEIDON system BGC fixed platforms (yellow dots, see Sect. ), glider endurance line (red line, see Sect. 4.6) and Ferrybox (yellow line, see Sect. 4.4).** Inset map shows location within the Mediterranean Sea (red square), location of other BGC fixed platforms (from www.oceansites.org) and E1-M3A spatial footprint (green area) for Chl-a using satellite observations (redrawn after Henson et al., 2016). The E1-M3A location includes in addition to the BGC fixed platform, an ADCP and sediment traps.

[Figure]

**Figure 2: Platforms of the Cretan Sea biogeochemical-ecosystem observatory.**

[Figure]

**Figure 3: Coastal (left) and open sea (right) fixed platforms configuration with scheme of payload**

[Figure]

**Figure 4: Temperature recordings at the E1-M3A buoy from 2007 to 2017.**

[Figure]

[Figure]

**Figure 53: Periods of operation of the different platforms located in the open Cretan Sea (historical metadata within a circle of 20 nautical miles radius around the position of E1-M3A; exclusion of metadata made within 10 nautical miles from a coast). Before 2010 the metadata listed may not be exhaustive. CTD casts and net tows were made from the surface to the depth shown in the figure. Ferrybox entry point is located at 3 m depth. BGC: $O_2$ and/or fluorescence sensors. CT: conductivity and temperature.**

[Figure]

[Figure]

**Figure 6: Vertical4: Total Chl-a vertical distribution of temperature, salinity (CTD casts) and total Chl-a at the POSEIDON E1-M3A location from 2010 to 2017 (fluorometric analysis of seawater from bottle sampling) from 2010 to 2017 at the POSEIDON E1-M3A location.).**

[Figure]

**Figure 75: Backscatter coefficient Sv from the 75 kHz ADCP placed at the POSEIDON E1-M3A location. Hand-drawn trails are attributed to different groups of zooplanktonic and micro-nectonic organisms (from Potiris et al., this issue).**

[Figure]

[Figure]

**Figure 8: Comparison of POSEIDON Ferrybox's fluorescence data with satellite ocean colour data over a selected period (October 2017 to January 2018). FB data are separated depending on the direction of the vessel (green lines: southward towards Heraklion, red lines: northward towards Piraeus). Satellite data were obtained from the OCEANCOLOUR_MED_CHL_L3_REP_OBSERVATIONS_009_073 product available at**

http://marine.copernicus.eu/services-portfolio/access-to-products/Figure 6: POSEIDON Ferrybox's fluorescence recordings on 01/07/2012.

[Figure]

**Figure 9: Sea-surface height (top panel) derived from satellite altimetry in the Cretan Sea in November 2017, in comparison to the vertical distribution of temperature (middle panel) and salinity (bottom panel) obtained by the glider along a west-east transect (red dotted line in top panel).**

[Figure]

**Figure 107: Time and space resolution of biogeochemicalbiochemical data acquisition by the different platforms of the observatory. The list of variablesparameters can be found in tables 1 to 3. Space resolution is vertical except in the case of Ferrybox. Carbonate: pH, pCO₂ or C_T&A_T, Other chem: other chemical variablesparameters, Sed trap: sediment trap, Phyto & protozoo: phytoplankton and protozoans; zoo: metazoans (collected with nets), Zoo migr: ADCP backscatter data for zooplankton migration.**

[Figure]

**Figure 118: Left: Hydraulic winch positioned on a small RIB allowing casts >1000 m (Pettas et al., 2015); Right: Temperature and deployment speed against depth from a CTD cast made using the hydraulic winch. Deployment speed decrease above 150m was made in order for the CTD to respond better to rapid environment changes like thermocline.**

[Figure]

**Figure 12: ERSEM model's food web structure (left) and correspondence between model variables and measured _variables_ at the Cretan Sea _Observatory_ (right).**

---

## Referee Report (RR1)

Comments to the Second version of the manuscript: "An integrated open-coastal biogeochemistry, ecosystem and biodiversity observatory of the Eastern Mediterranean. The Cretan Sea component of POSEIDON system" from Petihakis et al.

The manuscript has been improved by the authors following the suggestions and correction of the three referees. I suggest just a few changes before being published.

General Comments:

Although the authors claimed I can't see the reduction of the 40% of section 2, which from my point of view it is still too long. It leads to diverting the reader's attention from the true content of the article.

Other parts of the ms have been modified and/or reduced but not well integrated leading to a fragmentation of the reading. One possibility, which I propose, is to unify points 4.1 and 4.3 which refer directly to the observations made on the fixed platform (by slightly modifying the title), regardless of the type of parameter, so as to incorporate everything into one.

Minor comments:

Page 5 Line 35: "In 2017, an extended upgrade and renewal of POSEIDON buoy network monitoring parts and components and their supporting hardware was realized with the implementation of an integrated marine monitoring program funded by the EEA Financial Mechanism 2009-2014. ….. *How was done this extended upgrade in 2017 with 2009-2014 funds?*

Page 7 Line 19: replace *Lagrangian*

Page 7 Line 33: *S*outhern Adriatic

Page 9 Line 15-17: non-easy to read, please change in: "The system was reactivated in 2012 for a few years until the end of 2014. Recently, since mid-207 it has been running continuously with an upgrade to also provide also $O_2$, $CO_2$ and pH measurements"

Page 9 line 22: change displays for depicts

Page 9 line 23: …. That need *further* investigation

Page 11 line 5: "Specialized personnel is the *main* component for smooth *and* continuous functioning of any multiplatform-multidisciplinary observatory"

Page 11 line 13: replace of with in

Page 11 line 28-29: add "proved *to be a* powerful"

Page 14 line 31: in-situ

---

## Author Response (AR2)

**Replies to Topic Editor**

We would like to thank also the topic editor for the additional comments that helped further improve the manuscript. We give our response to his comments:

Comments to the Second version of the manuscript: "An integrated open-coastal biogeochemistry, ecosystem and biodiversity observatory of the Eastern Mediterranean. The Cretan Sea component of POSEIDON system" from Petihakis et al.

The manuscript has been improved by the authors following the suggestions and correction of the three referees. I suggest just a few changes before being published.

General Comments:
Although the authors claimed I can't see the reduction of the 40% of section 2, which from my point of view it is still too long. It leads to diverting the reader's attention from the true content of the article.

Reply 1:
The information provided concerning the reduction of Section 2 by approximately 40% was correct (i.e. 39% from 2594 words in first version to 1584 words in the second version). We have further reduced the length of this section to approximately 50% (i.e. 49% from 2594 words in first version to 1330 words in the current version). We consider a further reduction will have a negative impact, as it will not allow to keep showing the particularities of the Cretan Sea (similarities, dissimilarities within the wider Eastern and entire Mediterranean Sea) in relation to the importance of implementing and sustaining an integrated OS in this area, and the benefits to the community beyond the specific area.

Other parts of the ms have been modified and/or reduced but not well integrated leading to a fragmentation of the reading. One possibility, which I propose, is to unify points 4.1 and 4.3 which refer directly to the observations made on the fixed platform (by slightly modifying the title), regardless of the type of parameter, so as to incorporate everything into one.

Reply 2: We have merged 4.1. and 4.3 to a single section with modified title, as suggested.

Minor comments:
Page 5 Line 35: "In 2017, an extended upgrade and renewal of POSEIDON buoy network monitoring parts and components and their supporting hardware was realized with the implementation of an integrated marine monitoring program funded by the EEA Financial Mechanism 2009-2014. ..... *How was done this extended upgrade in 2017 with 2009-2014 funds?*

Reply 3 : The call's name was EEA Financial Mechanism 2009-2014 and it was implemented in 2017 (due to several administrative delays).

Page 7 Line 19: replace *Lagrangian*
Page 7 Line 33: *S*outhern Adriatic
Page 9 Line 15-17: non-easy to read, please change in: "The system was reactivated in 2012 for a few years until the end of 2014. Recently, since mid-207 it has been running continuously with an upgrade to also provide also $O_2$, $CO_2$ and pH measurements"
Page 9 line 22: change displays for depicts
Page 9 line 23: .... That need *further* investigation
Page 11 line 5: "Specialized personnel is the *main* component for smooth *and* continuous

functioning of any multiplatform-multidisciplinary observatory"
Page 11 line 13: replace of with in
Page 11 line 28-29: add "proved *to be a* powerful"
Page 14 line 31: in-situ

Reply 4 : All above minor comments were made.

The font size in figure 4 is too small and we recommend enlarging it to make the graph more visible.

Reply 5 : The font size in figure 4 was enlarged.

Besides to the above adjustments requested by the Topic Editor, we checked again the manuscript for typos, missing co-authors, terminology and updates of data in tables.
Finally figure 7 was updated based on the latest (accepted) version of the manuscript by Potiris et al. (in the same issue).

[revised manuscript text omitted]

The Cretan Sea is an area of intermediate and/or deep-water formation, dominated by multiple scale circulation patterns and intense mesoscale variability (e.g. Georgopoulos et al., 2000). Such areas of water formation are key locations for monitoring of the Mediterranean biogeochemical functioning (Malanotte-Rizzoli et al., 2014).

In the late 1980s - early 1990s, the Cretan Sea has become the major contributor of deep/bottom, warmer, more saline water to the Eastern Mediterranean (review by Laskaratos et al., 1999 and references thereinTheocharis et al., 2014). This transition known as the Eastern Mediterranean Transient (EMT) alternated dramatically the physical and biogeochemical properties at the intermediate and deep layers of the whole basin (Roether et al., 2007, among others). Data collected from different platforms in the Cretan Sea during the 2000s present evidence of gradually increasing salinity in the intermediate and deep intermediate layers after the middle of the decade. From late 2000s to early 2010s, dense water formation conditions have been identified showing that the basin is  slowly returning to the state of the late pre-EMT period (Velaoras et al., 2014; Cardin et al., 2015).

The deep-water mass formation in the Cretan Sea has the potential to transfer $CO_2$ into the deep layers. However, there is a sparseness of carbonate system data, i.e. total inorganic carbon ($C_T$), total alkalinity ($A_T$), pH and pCO$_2$ in the wider Eastern Mediterraneanto understand its role as source or sink of $CO_2$ (González-Dávila et al. 2016; Sisma-Ventura et al., 2016)

and the existing $A_T$-S relationships for the Mediterranean do not reproduce efficiently the $A_T$ levels observed in the Aegean Sea (Krasakopoulou et al., 2015).

5

The significant variability in the circulation patterns (e.g. Korres et al., 2014) and the strong coupling with the biogeochemical processes in the Cretan Sea (Tselepides and Polychronaki, 2000) has made evident that sparse spatial and temporal observations are prone to misrepresentation of the underlying dynamics.

10 **2.2 Oligotrophy, primary production, plankton stock and biodiversity hot spots**

The Eastern Mediterranean Basin (including the Cretan Sea) is considered to be an ultra-oligotrophic system, in terms of both Chl-a concentration (D'Ortenzio and Ribera d'Alcala, 2009) and primary productivity  (Siokou-Frangou et al., 2010 and references therein).

15  The dominance of the multivorous food web  in the Cretan Sea (Siokou-Frangou et al., 2002), is a dissimilarity with most areas of the Mediterranean Sea where the microbial food web generally dominates (Siokou-Frangou et al., 2010 and references

20 therein).

The Mediterranean Sea also constitutes a hot spot of biodiversity with a uniquely high percentage of endemic species (Coll et al., 2010 and references therein). However, biodiversity studies are still limited in the Mediterranean both for benthos (Danovaro et al., 2010) and plankton  (Siokou-Frangou et al., 2010). In addition, the Eastern Mediterranean is more

25 subject to change by the invasion of alien species in combination with warming (Coll et al., 2010) . All the above make clear the interest of studying the biodiversity of the open Cretan Sea.

**2.3 Biological pump efficiency at mid and deep waters**

The inhabitants of the deep sea play an important role in determining the depths to which carbon is

30 exported, a role mainly played by microbes and zooplankton (review by Turner, 2015). The lack of data from midwater depths severely limits our ability to quantify the efficiency of the biological pump (Robinson et al., 2010). In the  eastern part of Mediterranean less is known on deep living zooplankton  (review by Saiz et al., 2014).

In the Cretan Sea, deep water mesozooplankton has occasionally been studied (Siokou et al., 2013) and

35 the vertical flux of zooplankton faecal pellets quantified (Wassman et al., 2000). In this area, macrozooplankton vertical migration appears to occur at diel and seasonal scale down to 500 m  (Potiris et al., this issue). Since zooplankton vertical migration may constitute an important active vertical flux

increasing the biological pump's efficiency (review by Frangoulis et al., 2005), its role in the Cretan Sea pump needs exploration.

**2.4 Basin to global anthropogenic impact**

Although it is clear that human activities have modified the biogeochemical cycles of nutrients
5  (Galloway et al., 2008), the understanding of the marine ecosystems' response to these variations is limited. The Eastern Mediterranean is characterized by an anomalous N:P ratio ranging from 25 to 28, significantly higher than the normal oceanic Redfield ratio (16:1) (Kress, 2003). In this basin, atmospheric deposition is believed to be the main source of nutrients in the euphotic zone of the open sea, other than the vertical mixing of water during winter (Christodoulaki et al., 2013 and
10 references therein), but the exact contribution to the balance of nutrients and the resulting impact on the productivity remains uncertain (Duce et al., 2008).

 Under global warming,  a stronger stratification of the water column (Barale et al., 2008)  could
15 increasing the importance of the external sources of nutrients like atmospheric deposition. However, thermal instability in the atmosphere,  could lead to strong convective events in the atmosphere that could also increase the vertical mixing in the water column, bringing up  nutrients  (Christodoulaki et al., 2016).

Besides atmospheric deposition, a large part of the Mediterranean coasts host areas with coastal water
20 contamination occurring by pollutants (EEA, 2006) and changes on the quality and quantity of river inputs (Ludwig et al., 2009).

[revised manuscript text omitted]

Since 2017 two more POSEIDON buoys outside the Cretan Sea, also provide biogeochemical data ($O_2$, Chl-a). These two similar to E1-M3A, Oceanor Wavescan buoys are deployed in the N. Aegean and Ionian waters. The buoy placed between Athos peninsula and the island of Lemnos (AB) (Fig. 1), monitors an area affected (e.g. with increased productivity) by the Black Sea water entering the North Aegean through the Dardanelles straits, which plays a significant role modulating the thermohaline characteristics and dynamics of the whole Aegean Sea, including the Cretan Sea (e.g. Korres et al., 2014). The Pylos site (PB) in the SE Ionian is a crossroad where intermediate and deep-water masses meet. The site is located on the pathway of the Aegean Sea dense water that travels to the north along the western coast of Greece. These three buoys (E1-M3A, AB and PB) providing meteorological, physical, and biogeochemical data in different areas, have allowed comparison of trends of physical variables (such as temporal increasing trend in temperature, see Fig. 4) and may allow the future comparison of biogeochemical variables in different levels of oligotrophy.

**4.2 Water column sampling and sediment traps (R/V cruises). From occasional high spatial resolution snapshots to sustained temporal coverage**

[revised manuscript text omitted]
 (e.g. dinoflagellates >20 μm and <20 μm) and/or trophic functioning (e.g. autotrophic and heterotrophic nanoflagellates). Mixotrophs data field data were the most particular case, since they were split in four subgroups, creating an autotrophic and a heterotrophic subgroup to express mixotrophy (not represented in the model), which were subdivided again based on cell size (>20 μm and <20 μm). A model application using such field data grouping/splitting for validation can be found in Tsiaras et al. (2017). Planktonologists and the modellers of the observatory agreed that there is no rule of thumb, and model-field data correspondences should be adjusted according to the season and region.

**6 Metadata and data handling**

The POSEIDON database is set to include BGC and BGC-associated variables, either remotely sensed (pH, pCO$_2$, Chl-a, O$_2$, meteorological, T, S), or obtained from in -situ sampling (Chl-a, O$_2$, nutrients, plankton stock). The processing and the Quality Control procedures for the data collected from all the POSEIDON stations comply with the Copernicus Marine Environment Monitoring Service (CMEMS) In Ssitu Thematic Assembly Center (INS TAC) procedures, as HCMR is the regional data distribution node for the Mediterranean Sea. A number of quality control procedures for the validity of the data and a series of metadata correctness tests are applied before the release of the relevant data files. The data quality control process includes different routines for Near Real-Time (NRT) products and the Delayed Mode/Reprocessed Products.

The Near real Time quality control consists of a set of automatic tests according to the EuroGOOS Data Management, Exchange and Quality Working Group (DATA-MEQ) recommendations (http://eurogoos.eu/data-management-exchange-quality-working-group-data-meq/). These procedures are defined by variable, elaborated

in coherence with international agreements, in particular those adopted within SeaDataNet project (https://www.seadatanet.org/Standards/Data-Quality-Control) and they are applied by all the regional nodes of CMEMS  INS TAC on the NRT products in order to assure a minimum level of quality. Detailed information on the applied procedures can be found in the CMEMS INS_TAC QC procedure manuals for temperature and salinity, currents and sea level (http://dx.doi.org/10.13155/36230), waves (http://doi.org/10.13155/46607), Chl-a (fluorescence), dissolved oxygen and nutrients (http://doi.org/10.13155/36232-).

During the Delayed Mode/Reprocessed data analysis, procedures assessing the consistency of the data over a period of time are applied to the time series. The scientific validation includes statistical tests to check the consistency of the observations and climatological tests to highlight suspicious data that could not be detected by the automatic Quality Control processes. The resulted outliers are reassessed through visual inspection, a procedure that has an increased level of complexity on its implementation and highly relies on scientific expertise. Delayed Mode/Reprocessed products are available for temperature, salinity and waves, while the relevant product for Chl-a (fluorescence) and dissolved oxygen is under development.

Data can be visualized through the POSEIDON web-site (fixed platforms, Ferrybox) and the MonGOOS data portal (http://www.mongoos.eu/data-center, all platforms except sediment traps, ADCP), while the data are freely available to the public, the stakeholders and the scientific community, acknowledging the EC INSPIRE directive, in order to enable easy access to data and their reuse.

**7 Management, governance and sustainable development**

POSEIDON, and thus its subsystem of the Cretan Sea Observatory, has been managed up to now at institutional level by the HCMR Operational Oceanography Unit (i.e. POSEIDON team) which is subdivided in four components, three headed by scientists (observing, forecasting/modelling, data center) and one headed by an engineer (technical component). Within this management, the choice of infrastructures, sensors and thus BGC variables has been mainly driven by the participation in EU projects (e.g. JERICO, FixO3, JERICO-NEXT) which follow EU and International priorities. The recent emphasis given to biogeochemistry has been mainly the outcome of participation to JERICO and JERICO-NEXT. The latest project will allow to the Cretan Sea Observatory to integrate in the nearby future new biogeochemical variables by following the recent developments in sensors observing carbonate system variables, pollutants, phytoplankton and microbial diversity, toxic algae, etc.

In the near future the management/governance will move to a national level, as in 2018 the Hellenic Integrated Marine and Inland Observing Forecasting and Offshore Technology Systems Observing and Forecasting System (HIMIOFOTS), a national research marine observing infrastructure has been initiated. Within the HIMIOFOTS network a national management/governance is planned subdivided in (i) coordination team, (ii) operational and development team, (iii) scientific/technological committee and (iv) advisory team for infrastructure users. These boards will supervise the execution of the infrastructure's strategic plan, the scientific excellence, the technological development/innovation, the long-term sustainability, the good access to the infrastructure by national and international scientists, the outreach activities and the participation in large research international infrastructure networks.

A critical issue for any observatory is its sustainability. A continuous funding which will allow not only the day-to-day operations, but also the upgrade to the current state of the art technology is crucial. Unfortunately, in most cases marine observatories in Europe are developed through intermittent funding and national incentives. Likewise, the fixed platform of the Cretan Sea's observatory was founded through the FP6 EU Mediterranean Forecasting System Pilot Project (MFSPP) followed by the POSEIDON project and the EFTA funds. Furthermore, some activities and developments have been funded in the framework of both EU and national projects (research and infrastructural), while in 2018 the observatory became part of the HIMIOFOTS, a research infrastructure part of the Greek Research Infrastructure road map.

In periods when funds are limited, it is important to have and maintain a baseline via prioritisation of the variables observed and platforms used. Such a plan must take into consideration, among other, the existing historical data sets, the international priorities and efforts and the specific scientific questions in the wider area (Eastern Mediterranean).

Despite some periods of low support, the multivariable-multiplatform approach and the resulting scientific production (>80 peer reviewed publications, >170 conference presentations, 7 PhD thesis) allowed the participation to various targeted research projects and thus the provision of funds through multiple sources. In addition, the long experience in the entire chain of operations and the particular conditions of the Eastern Mediterranean make the observatory an excellent test bed both for new technology and sampling methods.

**8    Future scopes, expansion and vision**

The future of the observatory is presented in a short-term strategy and a long-term vision.

1. The short-term strategy of the biogeochemical observatory follows the expansion vision of POSEIDON system which considers recommendations, guidelines and priorities defined in the national Research Infrastructure road map of observing systems (HIMIOFOTS), review papers (e.g. Claustre et al., 2010), EU goals directives (MSFD, H2020), and visions of European (EMB and EuroGOOS, e.g. EGMRI, 2013) and International coordinating bodies (Global Ocean Observing system - GOOS, and Global Climate Observing System - GCOS).

A main short-term goal is attaining a NRT character for the biogeochemical variables together with a further expansion of the recorded variables, with a greater focus in air-sea interaction. Based on the key variables recommended by the EU (EGMRI, 2013), priority is currently given to further integrate sensors of $O_2$, $CO_2$, pH and fluorescence (Chl-a). Nutrient sensing is expected to be the next to follow, although the low concentrations found in the Eastern Mediterranean constitute a strong technological challenge. HCMR plans to expand the ability to host such biogeochemical sensors (with NRT capability) to more of the existing POSEIDON platforms (e.g. buoys, Bio-Argos, gliders, drifters, Ferrybox) beyond the Cretan Sea, as done in 2017 with the addition of biogeochemical sensors to the NE Aegean and SE Ionian Sea buoy. 
[revised manuscript text omitted]

| | Ferrybox | Thermo-Salinometer SBE 45 | ± 0.005 °C / ± 0.0005 S/m | 1min |
| | RV (CTD) | SBE 19+ OR SBE911 | ± 0.005 °C / ± 0.0005 S/m | 1 sec |
| | Glider | SBE GPCTD (Glider Payload CTD) | ± 0.002 °C / ± 0.0003 S/m | 30 sec |
| Fluorescence (Chl-a) | Buoys (E1-M3A, AB, PB) | Wetlabs FLNTU | 0.025 µg/L Chl | 180 min |
| | Ferrybox | Scufa II Turner Design | 0.02 µg/L Chl | 1 min |
| | RV (CTD) | WET Labs ECO-AFL/FL 9 or Chelsea Aqua 3 | 0.025 µg/L Chl | 1 sec |
| Dissolved Oxygen | Buoys (E1-M3A, AB, PB) | SBE43 / SBE 63/ Aanderaa Optode | ± 2% / ± 2%/ ± 5% | 180 min |
| | Ferrybox | Aanderaa Ooptode | ± 5% | 1 min |
| | Bio-Argos | Aanderaa Ooptode | ± 5% | 1 min |
| | RV (CTD) | SBE 43 | ± 2% | 1 sec |
| | Glider | SBE 43F | ± 2% | 30 sec |
| Turbidity | RV (CTD) | WET Labs ECO FLNTU | 0.013 NTU | 1 sec |
| PAR/Irradiance | RV (CTD) | Biospherical/Licor | N/A | 1 sec |
| pH | Buoy (E1-M3A) | Sensor LabpH | ± 0.005 pH units | 180 min |
| | Ferrybox | Meinsberg probe | ± 0.3 pH units | 1 min |
| pCO$_2$ | Ferrybox | ControsCO2 | ± 0.5% | 1 min |
| | Buoy (E1-M3A) | Pro-Oceanus | 2 ppm | 180 min |
| ADCP (backscattering) | Buoy (E1-M3A) | Teledyne RDI 75kHz | ± 1% | 180 min |

**Table 2: Variables measured from discrete bottle and net samples at high frequency (monthly). Method ranking from Lorenzoni and Benway (2013).**

| Variable | Platform | Analytical method | Method Ranking | Accuracy |
|---|---|---|---|---|
| $NO_3+NO_2$, $Si(OH)_4$ | E1-M3A, HCB | Manual Spectrophotometric | Acceptable | <3% <4% |
| $PO_4$ | E1-M3A, HCB | Magnesium-induced co-precipitation | Best | <2% |
| Total Chl-a | E1-M3A, HCB | Fluorescence and HPLC | Best | NA |
| Other Phytopigments | E1-M3A, HCB | HPLC | Best | NA |
| Viruses and Bacteria | E1-M3A | Flow cytometry | Best | NA |
| Picophytoplankton | E1-M3A | Flow cytometry | Best | NA |
| Nanophytoplankton | E1-M3A, HCB | Microscopy (UV + blue light excitation) | Best | NA |
| Other nanoplankton | E1-M3A, HCB | Inverted microscopy | NA | NA |
| Microphytoplankton | E1-M3A, HCB | Inverted microscopy | Best | NA |
| Ciliates | E1-M3A | Inverted epifluorescence microscope (blue light excitation) | NA | NA |
| Zooplankton | E1-M3A, HCB | 45μm and 200 μm nets, Scanning & Image analysis | Best | NA |

**Table 3: Variables measured from discrete bottle samples at low frequency (6 to 24 months) and from sediment traps (integrating 15 days). Method ranking from Lorenzoni and Benway (2013). +traps : measurement from both water column samples and from particulate matter in sediment traps. FB: Ferrybox.**

| Variable | Platform | Analytical method | Method Ranking | Accuracy |
|---|---|---|---|---|
| Dis. Oxygen | E1-M3A, HCB | Winkler (against CTD sensor) | Best | <0.02 mL/L |
| $A_T$ | E1-M3A, FB | Potentiometric titration (Closed cell) | Best | <0.1% |
| DIC | E1-M3A, FB | Coulometric determination | Best | <0.1% |
| TEP | E1-M3A | Colorimetric determination | NA | <11% |
| POC/ PN | E1-M3A (+ traps) | High Temperature Combustion via Elemental Analyzer | Good | <3.5% / <2.5% |
| DOC or TOC | E1-M3A | High Temperature Combustion | Best | <3% |
| TDN | E1-M3A | Persulfate Oxidation | Good | <5% |
| DOP | E1-M3A | Persulfate Oxidation | Best | <5% |
| Primary production | E1-M3A | 14C. Fractional day incubations scaled to daily rates | Acceptable | NA |
| Bacterial production | E1-M3A | $^3$H-labelled leucine method | Best | NA |

[Figure]

**Figure 1: Map of showing the location of the Cretan Sea (green ellipse) and all POSEIDON system BGC fixed platforms (yellow dots, see Sect. 4.1), glider endurance line (red line, see Sect. 4.5) and Ferrybox (yellow line, see Sect. 4.3). Inset map shows location within the Mediterranean Sea (red square), location of other BGC fixed platforms (from www.oceansites.org) and E1-M3A spatial footprint (green area) for Chl-a using satellite observations (redrawn after Henson et al., 2016). The E1-M3A location includes in addition to the BGC fixed platform, an ADCP and sediment traps.**

[Figure]

**Figure 2: Platforms of the Cretan Sea biogeochemical-ecosystem observatory.**

[Figure]

**Figure 3: Coastal (left) and open sea (right) fixed platforms configuration with scheme of payload**

[Figure]

[Figure]

**Figure 4: Temperature recordings at the E1-M3A buoy from 2007 to 2017.**

[Figure]

**Figure 5: Periods of operation of the different platforms located in the open Cretan Sea (historical metadata within a circle of 20 nautical miles radius around the position of E1-M3A; exclusion of metadata made within 10 nautical miles**

5 **from a coast). Before 2010 the metadata listed may not be exhaustive. CTD casts and net tows were made from the surface to the depth shown in the figure. Ferrybox entry point is located at 3 m depth. BGC: $O_2$ and/or fluorescence sensors. CT: conductivity and temperature.**

[Figure]

**Figure 6: Vertical distribution of temperature, salinity (CTD casts) and total Chl-a (fluorometric analysis of seawater from bottle sampling) from 2010 to 2017 at the POSEIDON E1-M3A location.**

[Figure]

5    **Figure 7: Backscatter coefficient Sv from the 75 kHz ADCP placed at the POSEIDON E1-M3A location. Hand-drawn trails are attributed to different groups of zooplanktonic and micro-nectonic organisms (from Potiris et al., this issue).**

[Figure]

**Figure 8: Comparison of POSEIDON Ferrybox's fluorescence data with satellite ocean colour data over a selected period (October 2017 to January 2018). FB data are separated depending on the direction of the vessel (green lines: southward towards Heraklion, red lines: northward towards Piraeus). Satellite data were obtained from the OCEANCOLOUR_MED_CHL_L3_REP_OBSERVATIONS_009_073 product available at http://marine.copernicus.eu/services-portfolio/access-to-products/**

[Figure]

**Figure 9: Sea-surface height (top panel) derived from satellite altimetry in the Cretan Sea in November 2017, in comparison to the vertical distribution of temperature (middle panel) and salinity (bottom panel) obtained by the glider along a west-east transect (red dotted line in top panel).**

[Figure]

**Figure 10: Time and space resolution of biogeochemical data acquisition by the different platforms of the observatory. The list of variables can be found in tables 1 to 3. Space resolution is vertical except in the case of Ferrybox. Carbonate: pH, pCO₂ or C_T&A_T, Other chem: other chemical variables, Sed trap: sediment trap, Phyto & protozoo: phytoplankton and protozoans; zoo: metazoans (collected with nets), Zoo migr: ADCP backscatter data for zooplankton migration.**

[Figure]

**Figure 11: Left: Hydraulic winch positioned on a small RIB allowing casts >1000 m (Pettas et al., 2015); Right: Temperature and deployment speed against depth from a CTD cast made using the hydraulic winch. Deployment speed decrease above 150m was made in order for the CTD to respond better to rapid environment changes like thermocline.**

[Figure]

**Figure 12: ERSEM model's food web structure (left) and correspondence between model variables and measured variables at the Cretan Sea Observatory (right)**